# Emergence of time persistence in a data-driven neural network model

Sebastien Wolf[1†], Guillaume Le Goc[2†], Georges Debrégeas[2‡], Simona Cocco[1‡], Rémi Monasson[1*‡]

[1]Laboratory of Physics of the Ecole Normale Supérieure, CNRS UMR 8023 & PSL Research, Sorbonne Université, Université de Paris, Paris, France; [2]Institut de Biologie Paris-Seine (IBPS), Laboratoire Jean Perrin, Sorbonne Université, CNRS, Paris, France

**Abstract** Establishing accurate as well as interpretable models of network activity is an open challenge in systems neuroscience. Here, we infer an energy-based model of the anterior rhomben-cephalic turning region (ARTR), a circuit that controls zebrafish swimming statistics, using functional recordings of the spontaneous activity of hundreds of neurons. Although our model is trained to reproduce the low-order statistics of the network activity at short time scales, its simulated dynamics quantitatively captures the slowly alternating activity of the ARTR. It further reproduces the modulation of this persistent dynamics by the water temperature and visual stimulation. Mathematical analysis of the model unveils a low-dimensional landscape-based representation of the ARTR activity, where the slow network dynamics reflects Arrhenius-like barriers crossings between metastable states. Our work thus shows how data-driven models built from large neural populations recordings can be reduced to low-dimensional functional models in order to reveal the fundamental mechanisms controlling the collective neuronal dynamics.

**\*For correspondence:**
remi.monasson@phys.ens.fr

[†]These authors contributed equally to this work
[‡]These authors also contributed equally to this work

**Competing interest:** The authors declare that no competing interests exist.

## Editor's evaluation

The authors show how high-dimensional neural signals can be reduced to low-dimensional models with variables that can be directly linked to behavior. The reduced model can account for long time scales of persistent activity that arise from transitions between metastable model states. The authors further show that the rate of these transitions is modulated by water temperature according to the classic Arrhenius law, although the results for different temperatures could not yet be unified into a single description based on real external temperature.

## Introduction

How computational capacities emerge from the collective neural dynamics within large circuits is a prominent question in neuroscience. Modeling efforts have long been based on top-down approaches, in which mathematical models are designed to replicate basic functions. Although they might be very fruitful from a conceptual viewpoint, these models are unable to accurately reproduce actual data and thus remain speculative. Recently, progress in large-scale recording and simulation techniques has led to the development of bottom-up approaches. Machine-learning models, trained on recorded activity, allow for the decoding or the prediction of neuronal activity and behavior (*Glaser et al., 2020*; *Pandarinath et al., 2018*). Unfortunately, the blackbox nature of these data-driven models often obscures their biological interpretation, for example, the identification of the relevant computational units (*Butts, 2019*). This calls for quantitative, yet interpretable approaches to illuminate the functions carried out by large neural populations and their neuronal substrate.

This work is an attempt to do so in the specific context of the anterior rhombencephalic turning region (ARTR), a circuit in the zebrafish larva that drives the saccadic dynamics and orchestrates the chaining of leftward/rightward swim bouts (*Ahrens et al., 2013*; *Dunn et al., 2016*; *Wolf et al., 2017*; *Ramirez and Aksay, 2021*; *Leyden et al., 2021*). The ARTR spontaneous activity exhibits temporal persistence, that is, the maintenance of activity patterns over long (~ 10 s) time scales. This functional feature is ubiquitous in the vertebrate brain. It plays an essential role in motor control, as best exemplified by the velocity position neural integrator, a circuit that integrates neural inputs and allows for a maintenance of the eye position after an ocular saccade (*Seung, 1996*; *Seung et al., 2000*; *Miri et al., 2011*). Temporal persistence is also central to action selection (*Wang, 2008*) and short-term memory storage (*Zaksas and Pasternak, 2006*; *Guo et al., 2017*). As isolated neurons generally display short relaxation times, neural persistence is thought to be an emergent property of recurrent circuit architectures (*Zylberberg and Strowbridge, 2017*). Since the 1970s, numerous mechanistic network models have been proposed that display persistent activity. They are designed such as to possess attractor states, that is, stable activity patterns toward which the network spontaneously converges.

Although attractor models are conceptually appealing, assessing their relevance in biological circuits remains challenging. To this aim, recent advances in machine learning combined with large-scale methods of neural recordings may offer a promising avenue. We hereafter focus on energy-based network models, trained to replicate low-order data statistics, such as the mean activities and pairwise correlations, through the maximum entropy principle (*Jaynes, 1957*). In neuroscience, such models have been successfully used to explain correlation structures in many areas, including the retina (*Schneidman et al., 2006*; *Cocco et al., 2009*; *Tkačik et al., 2015*), the cortex (*Tavoni et al., 2016*; *Tavoni et al., 2017*; *Nghiem et al., 2018*), and the hippocampus (*Meshulam et al., 2017*; *Posani et al., 2017*) of vertebrates, and the nervous system of *Caenorhabditis elegans* (*Chen et al., 2019*). These models are generative, that is, they can be used to produce synthetic activity on short time scales, but whether they can reproduce long-time dynamical features of the biological networks remains an open question.

Here, we first report on spontaneous activity recordings of the ARTR network using light-sheet functional imaging at various yet ethologically relevant temperatures. These data demonstrate that the water temperature controls the persistence time scale of the ARTR network, and that this modulation is in quantitative agreement with the thermal dependence of the swimming statistics. We then infer energy-based models from the calcium activity recordings and show how these data-driven models not only capture the characteristics and probabilities of occurrence of activity patterns, but also reproduce the observed thermal dependence of the persistent time scale. We further derive a mathematically tractable version of our energy-based model, called mean-field approximation, whose resolution provides a physical interpretation of the energy landscape, of the dynamical paths there in, and of their changes with temperature. We finally extend the model to incorporate visual stimulation and correctly reproduce the previously reported visually driven ARTR dynamics (*Wolf et al., 2017*). This work establishes the capacity of data-driven network inference to numerically emulate persistent dynamics and to unveil fundamental network features controlling such dynamics.

## Results

### The water temperature controls behavioral and neuronal persistence time scales in zebrafish larvae

In this first section, we report on functional recordings of the ARTR dynamics performed at various temperatures (18–33°C). We show that the persistent time scale that characterizes the ARTR's endogenous dynamics is thermally modulated. This dependence is reflected in the change in swimming statistics observed in freely swimming assays. We further characterize how the water temperature impacts the distribution of activity patterns.

#### ARTR endogeneous dynamics is thermally modulated

We used light-sheet functional imaging to record the ARTR activity in zebrafish larvae expressing a calcium reporter pan-neuronally (*Tg(elavl3:GCaMP6)*). The larvae, embedded in agarose, were placed in a water tank whose temperature was controlled in the range 18–33°C (see *Appendix 2—figure 1A*). ARTR neurons were identified using a combination of morphological and functional criteria, as

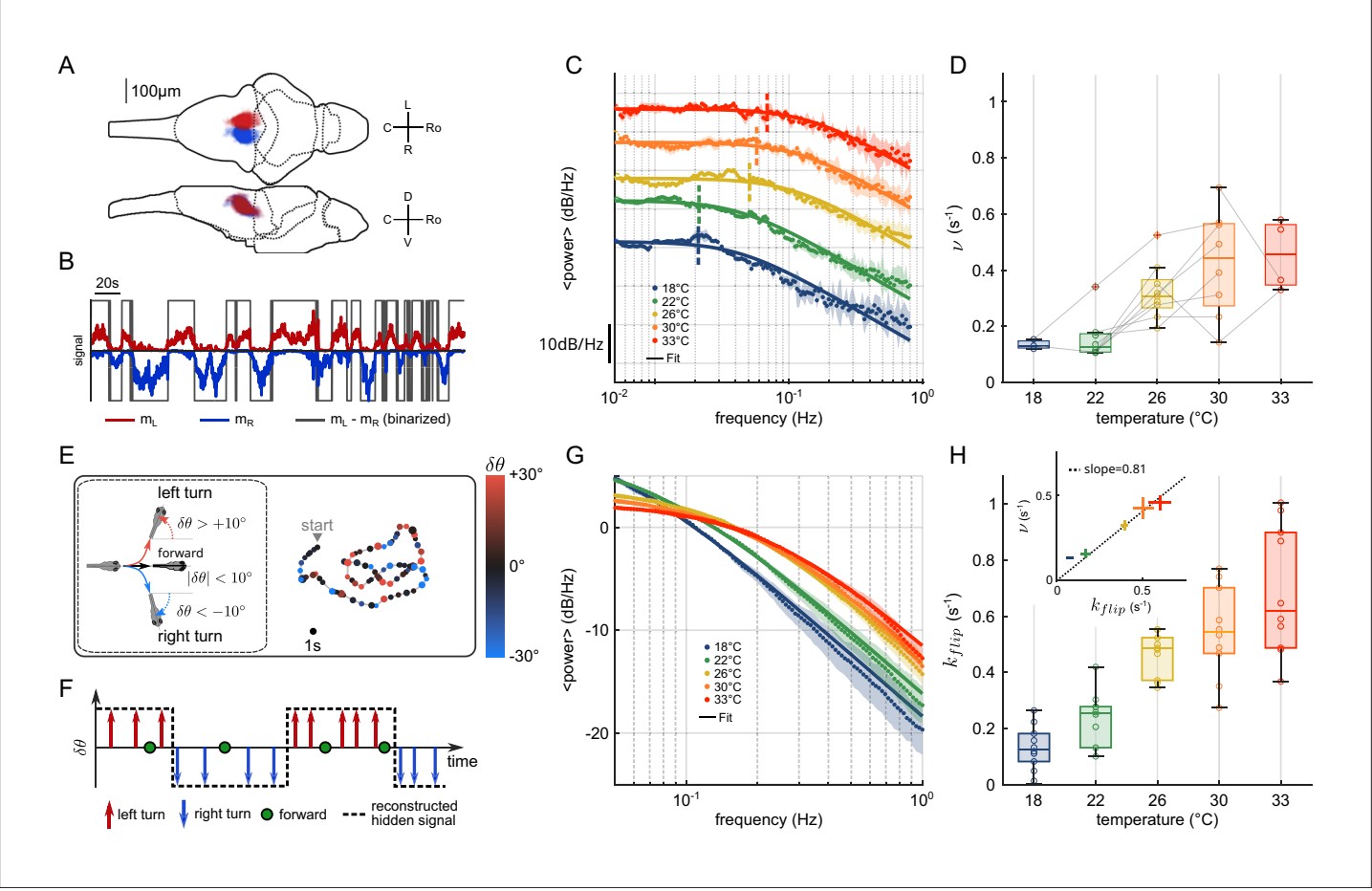

**Figure 1.** Temperature dependence of anterior rhombencephalic turning region (ARTR) dynamics and turn direction persistence. (**A**) Morphological organization of the ARTR showing all identified neurons from 13 fish recorded with light-sheet calcium imaging. (**B**) Example of ARTR binarized signal sign ($m_L - m_R$) (gray) along with the left ($m_L$, red) and right ($m_R$, blue) mean activities. (**C**) Averaged power spectra of the ARTR binarized signals for the five tested temperatures. The dotted vertical lines indicate the signal switching frequencies $\nu$ as extracted from the Lorentzian fit (solid lines). (**D**) Temperature dependence of $\nu$. The lines join data points obtained with the same larva. (**E**) Swimming patterns in zebrafish larvae. Swim bouts are categorized into forward and turn bouts based on the amplitude of the heading reorientation. Example trajectory: each dot corresponds to a swim bout; the color encodes the reorientation angle. (**F**) The bouts are discretized as left/forward/right bouts. The continuous binary signal represents the putative orientational state governing the chaining of the turn bouts. (**G**) Power spectra of the discretized orientational signal averaged over all animals for each temperature (dots). Each spectrum is fitted by a Lorentzian function (solid lines) from which we extract the switching rate $k_{flip}$. (**H**) Temperature dependence of $k_{flip}$. Inset: relationship between $k_{flip}$ (behavioral) and $\nu$ (neuronal) switching frequencies. Bar sizes represent SEM, and the dashed line is the linear fit.

detailed in *Wolf et al., 2017*. Their spatial organization is displayed in *Figure 1A*, for all recorded animals after morphological registration on a unique reference brain (145 ± 65 left neurons, 165 ± 69 right neurons, mean ± SD across 13 different fish, see *Appendix 2—table 1*). For each neuron, an approximate spike train $s(t)$ was inferred from the fluorescence signal using Bayesian deconvolution (*Tubiana et al., 2020*). A typical raster plot of the ARTR is shown in *Appendix 2—figure 1B* (recorded at 26°C), together with the mean signals of the left and right subcircuits, $m_{L,R}(t) = \frac{1}{N_{L,R}} \sum_{i \in L,R} s_i(t)$.

To analyze the thermal dependence of the ARTR dynamics, we extracted from these recordings a binarized ARTR signal, $\text{sign}(m_L(t) - m_R(t))$ (see *Figure 1B* and *Appendix 2—figure 1C* for example signals from the same fish at different temperatures). The average power spectra of these signals for the five tested temperatures (average of 3–8 animals for each temperature, see *Appendix 2—table 1*) are shown in *Figure 1C*. We used a Lorentzian fit to further extract the alternation frequency $\nu$ for each dataset (*Figure 1C*, solid lines). This frequency was found to increase with the temperature (*Figure 1D*). Although $\nu$ could significantly vary across specimen at a given temperature, for a given

animal, increasing the temperature induced an increase in the frequency in 87.5% of our recordings (28 out of 32 pairs of recordings).

In this analysis, we used the binarized ARTR activity to facilitate the comparison between behavioral and neural data, as described in the next section. However, the observed temperature dependence of the left-right alternation time scale was preserved when the spectra were computed from the ARTR activity, $m_L(t) - m_R(t)$ (see **Appendix 2—figure 1D**).

## Impact of the water temperature on the turn direction persistence in freely swimming larvae

It has previously been shown that the ARTR governs the selection of swim bout orientations: turn bouts are preferentially executed in the direction of the most active (right or left) ARTR subcircuit (**Dunn et al., 2016**; **Wolf et al., 2017**), such that $\text{sign}(m_L(t) - m_R(t))$ constitutes a robust predictor of the turning direction of the animal; see Figure 5 - figure supplement 2E in **Dunn et al., 2016**. Therefore, the temporal persistence of the ARTR dynamics is reflected in a turn direction persistence in the animal's swimming pattern, that is, the preferred chaining of similarly orientated turn bouts.

We thus sought to examine whether the thermal dependence of the ARTR endogenous dynamics could manifest itself in the animal navigational statistics. In order to do so, we used the results of a recent study (**Le Goc et al., 2021**), in which 5–7-day-old zebrafish larvae were video-monitored as they swam freely at constant and uniform temperature in the same thermal range (**Figure 1E**). We quantified the time scale of the turn direction persistence by assigning a discrete value to each turn bout: −1 for a right turn, +1 for a left turn (forward scouts were ignored). We then computed an orientational state signal continuously defined by the value of the last turn bout (**Figure 1F**). The power spectra of the resulting binary signals are shown in **Figure 1G** for various temperatures. We used a Lorentzian fit ('Materials and methods,' **Equation 6**) to extract, for each experiment, a frequency $k_{flip}$. This rate, which defines the probability of switching orientation per unit of time, systematically increases with the temperature, from 0.1 to 0.6 s$^{-1}$ (**Figure 1H**). Increasing the temperature thus leads to a progressive reduction of the turn direction persistence time. The inset plot in **Figure 1H** establishes that the left/right alternation rates extracted from behavioral and neuronal recordings are consistent across the entire temperature range (slope = 0.81, $R = 0.99$).

## ARTR activity maps are modulated by the temperature

We then investigated how the water temperature impacts the statistics of the ARTR activity defined by the mean activity of the left and right sub-populations, $m_L$ and $m_R$. The probability maps in the $(m_L, m_R)$ plane are shown in **Figure 2A** for two different temperatures, with the corresponding raster plots and time signals of the two subcircuits. At high temperature, the ARTR activity map is confined within an L-shaped region around $(m_L = 0, m_R = 0)$ and the circuit remains inactive for a large fraction of the time. Conversely, at lower temperature, the ARTR activity is characterized by long periods during which both circuits are active and shorter periods of inactivity. We quantified this thermal dependence of the activity distribution by computing the log-probability of the activity of either region of the ARTR at various temperatures (**Appendix 2—figure 2A**). The occupation rate of the inactive state ($m_{L,R} \sim 0$) increases with temperature, with a corresponding steeper decay of the probability distribution of the activity. Consistently, we found that the mean activities $m_L$ and $m_R$ decreased with temperature (**Appendix 2—figure 2B**). Such a dependence might reflect varying levels of temporal coherence in the activity of the ARTR with the temperature. In order to test this, we computed the Pearson correlation at various temperature but we saw no clear dependency of the average correlation across ipsilateral or contralateral pairs of neurons (**Appendix 2—figure 2C**).

Our analysis thus indicates that the water temperature modulates both the endogenous dynamics and the activity distribution of the ARTR. For both aspects, we noticed a large variability between animals at a given temperature. This is not unexpected as it parallels the intra- and inter-individual variability in the fish exploratory kinematics reported in **Le Goc et al., 2021**. Nevertheless, we observed a strong positive correlation between the persistence time and the mean activity across animals and trials for a given temperature (**Appendix 2—figure 2D** and 'Materials and methods'), indicating that both features of the ARTR may have a common drive.

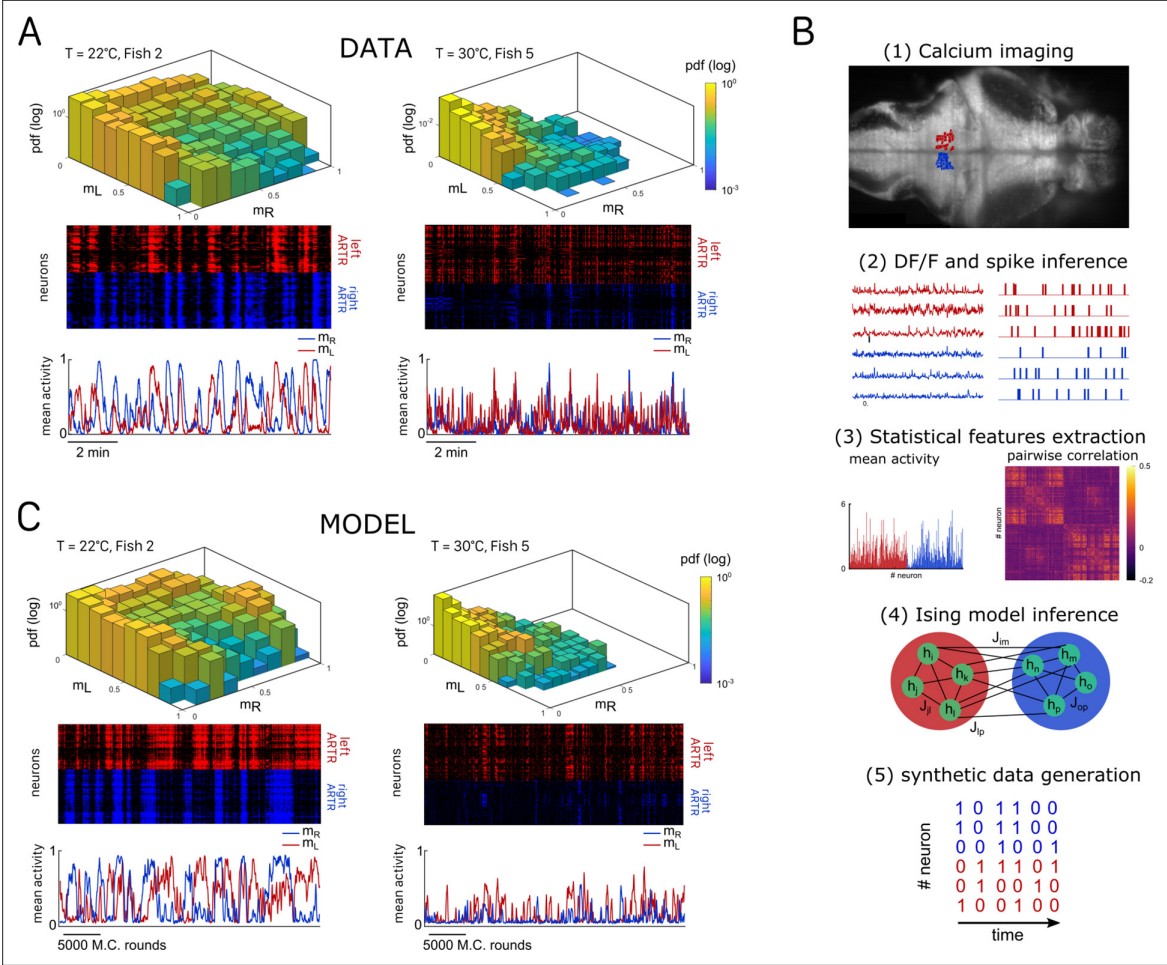

**Figure 2.** Ising models reproduce characteristic features of the recorded activity. (**A**) (Top) Probability densities $P(m_L, m_R)$, see Equation 2, of the activity state of the circuit (obtained from the spiking inference of the calcium data), in logarithmic scale, and for two different fish and water temperatures $T = 20$ and $T = 30°C$; Color encodes z-axis (same color bar for both). (Middle) 10-min-long raster plots of the activities of the left (red) and right (blue) subregions of the anterior rhombencephalic turning region (ARTR). (Bottom) Corresponding time traces of the mean activities $m_L$ and $m_R$. (**B**) Processing pipeline for the inference of the Ising model. We first extract from the recorded fluorescence signals approximate spike trains using a Bayesian deconvolution algorithm (BSD). The activity of each neuron is then '0' or '1.' We then compute the mean activity and the pairwise covariance of the data, from which we infer the parameters $h_i$ and $J_{ij}$ of the Ising model. Finally, we can generate raster plot of activity using Monte Carlo sampling. (**C**) Same as (**A**) for the two corresponding inferred Ising models. The raster plots correspond to Monte Carlo-sampled activity, showing slow alternance between periods of high activity in the L/R regions. Here we show only two examples of a qualitative experimental vs. synthetic signals comparison. We provide in the supplementary materials the same comparison for every recording.

## A data-driven energy-based model reproduces the statistics of the ARTR dynamics

Our aim was to reproduce the ARTR spontaneous activity using an energy-based data-driven network model. The inference pipeline, going from raw fluorescence data to the model, is summarized in *Figure 2B*. We first reconstructed an estimated spike train for each ARTR neuron using a deconvolution algorithm (*Tubiana et al., 2020*). We divided the recording window ($T_{rec} \sim 1200$ s for each session) in time bins whose width was set by the imaging frame rate ($dt = 100-300$ ms). Each dataset thus consisted of a series of snapshots $\mathbf{s}^{\mathbf{k}} = (s_1^k, \ldots, s_N^k)$ of the ARTR activity at times $k$, with $k = 1, \ldots, T_{rec}/dt$; here, $s_i^k = 1$ if cell $i$ is active or $s_i^k = 0$ if it is silent in time bin $k$.

We then computed the mean activities, $\langle s_i \rangle_{\text{data}}$, and the pairwise correlations, $\langle s_i s_j \rangle_{\text{data}}$, as the averages of, respectively, $s_i^k$ and $s_i^k s_j^k$ over all time bins $k$. We next inferred the least constrained model, according to the maximum entropy principle (*Jaynes, 1957*), that reproduced these quantities. This model, known as the Ising model in statistical mechanics (*Ma, 1985*) and probabilistic graphical model

in statistical inference (*Koller and Friedmann, 2009*), describes the probability distribution over all $2^N$ possible activity configurations **s**,

$$P\left(\mathbf{s}\right) = \tfrac{1}{Z} \, \exp\left(\sum_i h_i \, s_i + \sum_{i<j} J_{ij} \, s_i s_j\right) , \qquad (1)$$

where $Z$ is a normalization constant. The bias $h_i$ controls the intrinsic activity of neuron $i$, while the coupling parameters $J_{ij}$ account for the effect of the other neurons $j$ activity on neuron $i$ ('Materials and methods'). The set of parameters $\{h_i, J_{ij}\}$ were inferred using the Adaptative Cluster Expansion and the Boltzmann machine algorithms (*Cocco and Monasson, 2011*; *Barton and Cocco, 2013*; *Barton et al., 2016*). Notice that in *Equation 1*, the energy term in the parenthesis is not scaled by a thermal energy as in the Maxwell–Boltzmann statistics. We thus implicitly fix the model temperature to unity; of course, this model temperature has no relation with the water temperature $T$. Although the model was trained to reproduce the mean activities and pairwise correlations (see *Appendix 2— figure 3A–C* and 'Materials and methods' for fourfold cross-validation), it further captured higher-order statistical properties of the activity such as the probability that $K$ cells are active in a time bin (*Appendix 2—figure 3D*; *Schneidman et al., 2006*).

Once inferred, the Ising model can be used to generate synthetic activity configurations **s**. Here, we used a Monte Carlo (MC) algorithm to sample the probability distribution $P(\mathbf{s})$ in *Equation 1*. The algorithm starts from a random configuration of activity, then picks up uniformly at random a neuron index, say, $i$. The activity $s_i$ of neuron $i$ is then stochastically updated to 0 or to 1, with probabilities that depend on the current states $s_j$ of the other neurons (see *Equation 8* in 'Materials and methods' and code provided). The sampling procedure is iterated, ensuring convergence toward the distribution $P$ in *Equation 1*. This in silico MC dynamics is not supposed to reproduce any realistic neural dynamics, except for the locality in the activity configuration **s** space.

*Figure 2C* shows the synthetic activity maps and temporal traces of Ising models trained on the two same datasets as in *Figure 2A*. For these synthetic signals, we use MC rounds, that is, the number of MC steps divided by the total number of neurons ('Materials and methods'), as a proxy for time. Remarkably, although the Ising model is trained to reproduce the low-order statistics of the neuronal activity within a time bin only, the generated signals exhibit the main characteristics of the ARTR dynamics, that is, a slow alternation between the left and right subpopulations associated with long persistence times; see raster plots in *Figure 2C*.

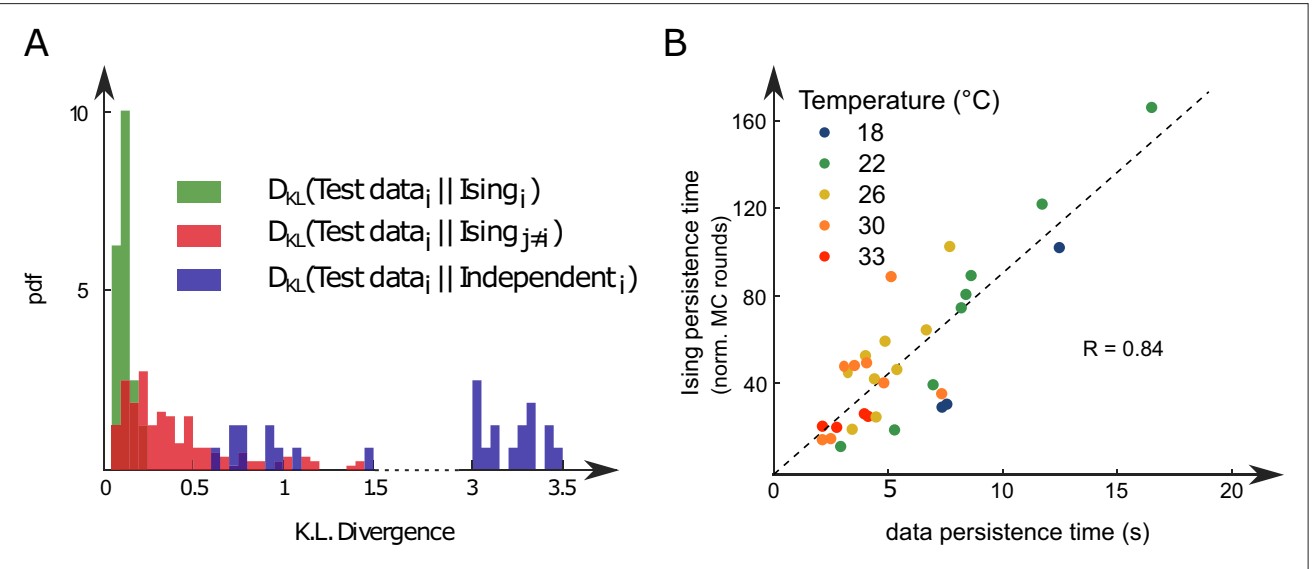

**Figure 3.** Comparison of model distributions and persistence times across fish and water temperatures. (**A**) Distribution of the Kullback–Leibler divergences between test datasets and their corresponding Ising models (green), between test datasets and Ising models trained on different datasets (red) and between test datasets and their corresponding independent models that assume no connections between neurons (dark blue). Note that each dataset is divided in a training set corresponding to 75% of the time bins chosen randomly and a test set comprising the remaining 25%. (**B**) Average persistence times in simulations vs. experiments. Each dot refers to one fish at one water temperature; colors encode temperature.

## Comparison of experimental and synthetic ARTR dynamics across recordings

We repeated the inference procedure described above for all our 32 recordings (carried out with $n = 13$ fish and 5 different water temperatures, see *Appendix 2—table 2*) and obtained the same number of sets of biases and couplings. We first compared the distributions of the left-right mean activity $m_L = \frac{1}{N_L} \sum_{i \in L} s_i$ and $m_R = \frac{1}{N_R} \sum_{i \in R} s_i$ extracted from the data and from the Ising model. In order to do so, we used the Kullback–Leibler (KL) divergence, a classical metrics of the dissimilarity between two probability distributions. The distribution of the KL divergences between the experimental test datasets (see 'Materials and methods') and their associated Ising models is shown in green in *Figure 3A*. The KL values were found to be much smaller than those obtained between experimental test datasets and Ising models trained from different recordings (red distribution). This result establishes that the Ising model quantitatively reproduces the ARTR activity distribution associated to each specimen and temperature.

This agreement crucially relies on the presence of inter-neuronal couplings in order to reproduce the pairwise correlations in the activity: a model with no connection (i.e., the independent model, see 'Materials and methods') fitted to reproduce the neural firing rates offers a very poor description of the data (see *Figure 3A* [dark blue distribution] and *Appendix 2—figure 3E–G*).

Finally, we examined to what extent the synthetic data could capture the neural persistence characteristics of the ARTR. The persistence times extracted from the data and from the MC simulations of the inferred models were found to be strongly correlated (*Figure 3B*, $R = 0.84$). The MC dynamics thus captures the inter-individual variability and temperature dependence of the ARTR persistent dynamics.

## Spatial organization and temperature dependence of the Ising inferred parameters

In all recordings, inferred ipsilateral couplings are found to be centered around a positive value (std = 0.12, mean = 0.062), while contralateral couplings are distributed around 0 (mean = –0.001, std = 0.10); see *Appendix 2—figure 4A–C*. Still, a significant fraction of these contralateral couplings are strongly negative. We illustrated this point by computing the fraction of neuronal pairs $(i, j)$ that are contralateral for each value of the coupling $J_{ij}$ or the Pearson correlation (*Appendix 2—figure 4D and E*). Large negative values of couplings or correlations systematically correspond to contralateral pairs of neurons, whereas large positive values correspond to ipsilateral pairs of neurons.

In addition, we found that the ipsilateral couplings $J_{ij}$ decay, on average, exponentially with the distance between neurons $i$ and $j$ (*Appendix 2—figure 4F*), in agreement with findings in other neural systems (*Posani et al., 2018*). Spatial structure is also present in contralateral couplings (*Appendix 2—figure 4G*). Biases display a wide distribution ranging from –8 to 0 (std = 1.1, mean = –4.1, *Appendix 2—figure 5A–C*), with no apparent spatial structure.

We next examined the dependency of the Ising model parameters on the water temperature. To do so, for each fish, we selected two different water temperatures, and the corresponding sets of inferred biases and couplings, $\{h_i, J_{ij}\}$. We then computed the Pearson correlation coefficient $R^2$ of the biases and of the coupling matrices at these two temperatures (inset of *Appendix 2—figure 6*). We saw no clear correlation between the model parameters at different temperatures, as shown by the distribution of $R^2$ computed across fish and across every temperatures (*Appendix 2—figure 6*).

## Mean-field study of the inferred model unveils the energy landscape underlying the ARTR dynamics

### Mean-field approximation to the data-driven graphical model

While our data-driven Ising model reproduces the dependence of the persistence time scale and activity distribution on the water temperature, why it does so remains unclear. To understand what features of the coupling and local bias parameters govern these network functional properties, we turn to mean-field theory. This powerful and mathematically tractable approximation scheme is commonly used in statistical physics to study systems with many strongly interacting components (*Ma, 1985*). In the present case, it amounts to deriving self-consistent equations for the mean activities $m_L$ and $m_R$ of the left and right ARTR subpopulations (*Figure 4A* and Appendix 1).

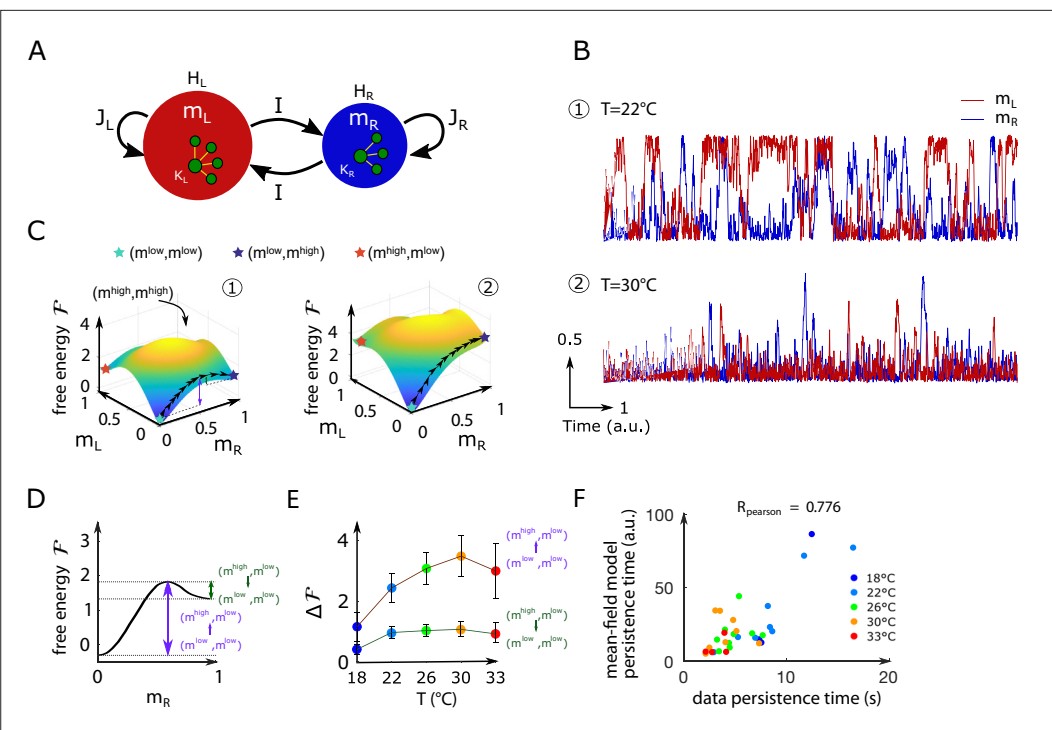

**Figure 4.** Mean-field approximation of the inferred Ising model. (**A**) Schematic view of the mean-field Ising model. (**B**) Examples of simulated $m_L$ and $m_R$ signals of the mean-field dynamical equations for two sets of parameters that correspond to fish ID #5 at two water temperatures (22°C and 30°C), see **Appendix 2—table 2**. (**C**) Free-energy landscapes in the ($m_L$,$m_R$) plane computed with the mean-field model. These data correspond to the same sets of parameters as in panel (**B**). Colored circles denote metastable states, and the line of black arrows indicates the optimal path between ($m^{low}$, $m^{low}$) and ($m^{low}$, $m^{high}$) states. (**D**) Schematic view of the free energy along the $m_R$ axes. The arrows denote the energy barriers $\Delta\mathcal{F}$ associated with the various transitions. The dark green arrow denotes $\Delta\mathcal{F}\left((m^{high},m^{low})\rightarrow(m^{low},m^{low})\right)$; the purple arrow denotes $\Delta\mathcal{F}\left((m^{low},m^{low})\rightarrow(m^{high},m^{low})\right)$. (**E**) Values of the free-energy barriers as a function of temperature. Error bars are standard error of the mean (32 recordings, n = 13 fish at 5 different water temperatures). (**F**) Persistence time of the mean-field anterior rhombencephalic turning region (ARTR) model for all fish and runs at different experimental temperatures. Each dot refers to one fish at one temperature; colors encode temperature.

Within mean-field theory, each neuron $i$ is subject to (i) a local bias $H$, (ii) an excitatory coupling $J > 0$ from the neurons in the ipsilateral region, and (iii) a weak coupling $I$ from the neurons in the contralateral side. These three parameters were set as the mean values of, respectively, the inferred biases $h_j$ and the inferred ipsilateral and contralateral interactions $J_{ij}$. In addition, we introduce an effective size $K$ of each region to take into account the fact that mean-field theory overestimates interactions by replacing them with their mean value. This effective number of neurons was chosen, in practice, to best match the results of the mean-field approach to the full Ising model predictions (see Appendix 1, **Appendix 2—table 2** and **Appendix 2—figure 7A–C**). It was substantially smaller than the number $N$ of recorded neurons. The selection method used to delineate the ARTR populations may yield different number of neurons in the $L$ and $R$ regions (see **Appendix 2—table 1**). This asymmetry was accounted for by allowing the parameters $H$, $J$, and $K$ defined above to take different values for the left and right sides.

Mean-field theory thus allowed us to reduce the data-driven Ising model, whose definition requires $\frac{1}{2}(N_L + N_R)(N_L + N_R + 1)$ parameters $\{h_i, J_{ij}\}$, to a model depending on seven parameters $(H_L, H_R, J_L, J_R, K_L, K_R, I)$ only (**Figure 4A**), whose values vary with the animal and the experimental conditions, for example, temperature (**Appendix 2—table 2**).

## Free energy and Langevin dynamics

The main outcome of the analytical treatment of the model is the derivation of the so-called free energy $\mathcal{F}(m_L, m_R)$ as a function of the average activities $m_L$ and $m_R$; see Appendix 1. The free energy

is a fundamental quantity as it controls the density of probability to observe an activation pattern $(m_L, m_R)$ through

$$P(m_L, m_R) \propto e^{-\mathcal{F}(m_L, m_R)} \tag{2}$$

Consequently, the lower the free energy $\mathcal{F}$, the higher the probability of the corresponding state $(m_L, m_R)$. In particular, the minima of the free energy define persistent states of activity in which the network can be transiently trapped.

The free energy landscape can be used to simulate dynamical trajectories in the activity space $(m_L, m_R)$. To do so, we consider a Langevin dynamics in which the two activities $m_L(t), m_R(t)$ evolve in time according to the stochastic differential equations,

$$\tau \frac{dm_L}{dt}(t) = -\frac{\partial \mathcal{F}}{\partial m_L}(m_L(t), m_R(t)) + \epsilon_L(t), \tag{3}$$

$$\tau \frac{dm_R}{dt}(t) = -\frac{\partial \mathcal{F}}{\partial m_R}(m_L(t), m_R(t)) + \epsilon_R(t), \tag{4}$$

where $\tau$ is a microscopic time scale, and $\epsilon_L(t), \epsilon_R(t)$ are white noise 'forces', $\langle \epsilon_L(t) \rangle = \langle \epsilon_R(t) \rangle = 0$, independent and delta-correlated in time: $\langle \epsilon_L(t)\epsilon_R(t') \rangle = 0$, $\langle \epsilon_L(t)\epsilon_L(t') \rangle = \langle \epsilon_R(t)\epsilon_R(t') \rangle = 2\,\delta(t - t')$. This Langevin dynamical process ensures that all activity configurations $(m_L, m_R)$ will be sampled in the course of time, with the expected probability as given by *Equation 2*.

*Figure 4B* shows the mean-field simulated dynamics of the left and right activities, $m_L$ and $m_R$, with the parameters corresponding to two Ising models at two different temperatures in *Figure 2C*. We observe, at low temperatures, transient periods of self-sustained activity (denoted by $m^{high}$) of one subcircuit, while the other has low activity ($m^{low}$) (see time trace 1 in *Figure 4B*). At high temperature, high activity in either (left or right) area can be reached only transiently (see trace 2 in *Figure 4B*). These time traces are qualitatively similar to the ones obtained with the full inferred Ising model and in the data (*Figure 2A and C*, bottom).

## Barriers in the free-energy landscape and dynamical paths between states

We show in *Figure 4C* the free-energy landscape in the $(m_L, m_R)$ plane for the same two conditions as in *Figure 4B*. The minimization conditions $\frac{\partial \mathcal{F}}{\partial m_L} = \frac{\partial \mathcal{F}}{\partial m_R} = 0$ provide two implicit equations over the activities $m_L^*, m_R^*$ corresponding to the preferred states. For most datasets, we found four local minima: the low-activity minimum $(m_L^*, m_R^*) = (m^{low}, m^{low})$, two asymmetric minima, $(m^{high}, m^{low})$ and $(m^{low}, m^{high})$, in which only one subregion is strongly active, and a state in which both regions are active, $(m^{high}, m^{high})$. The low-activity minimum $(m^{low}, m^{low})$ is the state of lowest free energy, hence with largest probability, while the high-activity state $(m^{high}, m^{high})$ has a much higher free energy and much lower probability. The free energies of the asymmetric minima $(m^{high}, m^{low})$ and $(m^{low}, m^{high})$ lie in between, and their values strongly vary with the temperature.

The Langevin dynamics defines the most likely paths (see 'Materials and methods') in the activity plane joining one preferred state to another, for example, from $(m^{high}, m^{low})$ to $(m^{low}, m^{high})$ as shown in *Figure 4C*. Along these optimal paths, the free energy $\mathcal{F}$ reaches local maxima, defining barriers to be overcome in order for the network to dynamically switchover (purple and green arrows in *Figure 4C*). The theory of activated processes stipulates that the average time to cross a barrier depends exponentially on its height $\Delta\mathcal{F}$:

$$t(\Delta\mathcal{F}) \sim \tau \times e^{\Delta\mathcal{F}}, \tag{5}$$

up to proportionality factors of the order of unity (*Langer, 1969*). Thus, the barrier $\Delta\mathcal{F}\left((m^{high}, m^{low}) \to (m^{low}, m^{low})\right)$ shown in dark green in *Figure 4D* controls the time needed for the ARTR to escape the state in which the left region is active while the right region is mostly silent, and to reach the all-low state. The barrier $\Delta\mathcal{F}\left((m^{low}, m^{low}) \to (m^{high}, m^{low})\right)$ shown in purple is related to the rising time from the low-low activity state to the state where the right region is active, and the left one is silent.

Within mean-field theory, we estimated the dependence in temperature of these barriers height (*Figure 4E* and *Appendix 2—figure 7D*) and of the associated persistence times (*Figure 4F*). While substantial variations from animal to animal were observed, we found that barriers for escaping the all-low state and switching to either $L, R$ region increase with the water temperature. As a consequence,

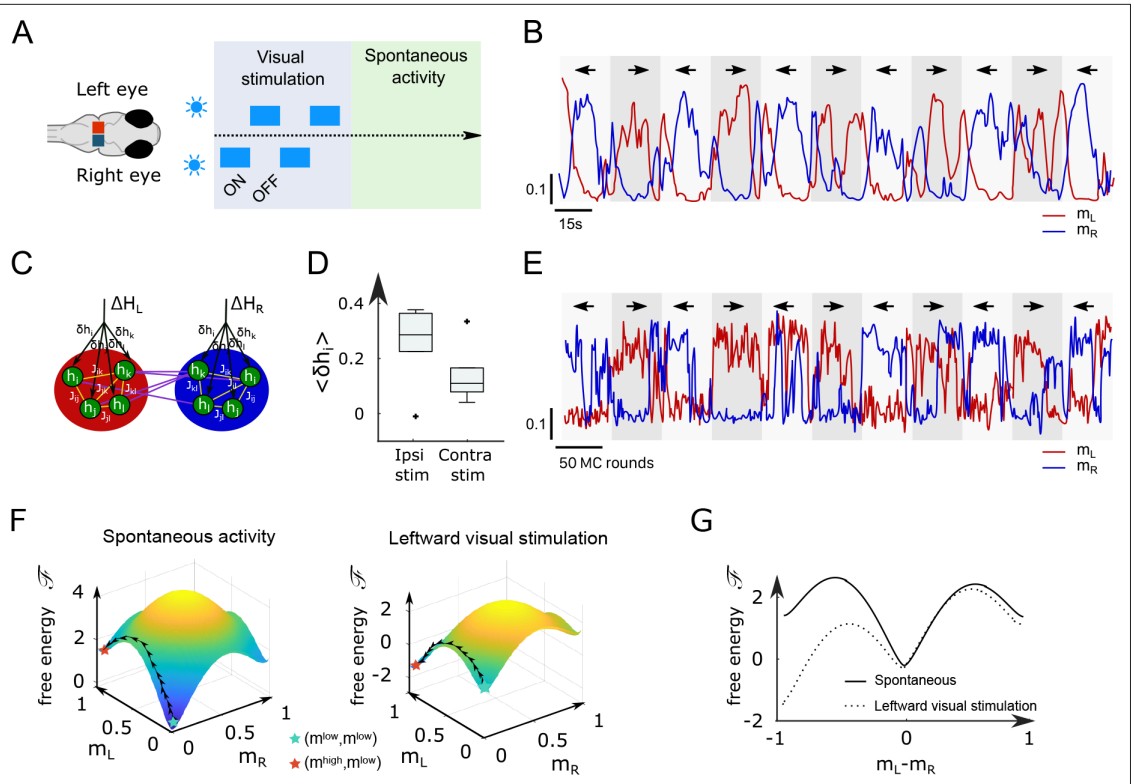

**Figure 5.** Modified Ising model captures the behavior of anterior rhombencephalic turning region (ARTR) under visual stimulation. (**A**) Scheme of the stimulation protocol. The left and right eyes are stimulated alternatively for periods of 15–30 s, after which a period of spontaneous (no stimulus) activity is acquired. (**B**) Example of the ARTR activity signals under alternated left–right visual stimulation. The small arrows indicate the direction of the stimulus. (**C**) Sketch of the modified Ising model, with additional biases $\delta h_i$ to account for the local visual inputs. (**D**) Values of the additional biases averaged over the ipsilateral and contralateral (with respect to the illuminated eye) neural populations. n = 6 fish. (**E**) Monte Carlo activity traces generated with the modified Ising model. (**F**) Free-energy landscapes computed with the mean-field theory during spontaneous (left panel) and stimulated (right panel) activity for an example fish. (**G**) Free-energy along the optimal path as a function of $m_L - m_R$ during spontaneous (plain line) and stimulated (dotted line) activity. The model is the same as in panel (**F**).

at high temperature, only the low–low activity state is accessible in practice to the system, and the mean activity remains low (see *Appendix 2—figure 2D*), with fluctuations within the low–low state. Conversely, at low water temperatures, barriers separating the low–low and the active high–low or low–high states are weaker, so the latter become accessible. As a first consequence, the mean activity is higher at low temperature (*Appendix 2—figure 2D*). Furthermore, the system remains trapped for some time in such an active state before switching to the other side, for example, from high–low to low–high. This is the origin of the longer persistence time observed at low temperature.

## Ising and mean-field models with modified biases capture the ARTR visually driven dynamics

While the analyses above focused on the spontaneous dynamics of the ARTR, our data-driven approach is also capable of explaining activity changes induced by external and time-varying inputs. In order to illustrate this capacity, we decided to reanalyze a series of experiments, reported in *Wolf et al., 2017*, in which we alternatively illuminated the left and right eyes of the larva, for periods of 15–30 s, while monitoring the activity of the ARTR (*Figure 5A*) with a two-photon light-sheet microscope. During and after each stimulation protocol, 855 s of spontaneous activity was recorded on $n = 6$ fish. We found that the ARTR activity could be driven by this alternating unilateral visual stimulation: the right side of the ARTR tended to activate when the right eye was stimulated and vice versa (*Figure 5B*).

To analyze these datasets, we first followed the approach described in *Figure 2B*, and inferred, for each fish, the sets of biases $h_i$ and interactions $J_{ij}$ using the spontaneous activity recording only (*Appendix 2—table 3*). In a *Appendix 2—table 2* second step, we exploited recordings of the visually

driven activity to infer additional biases $\delta h_i$ to the neurons, while keeping the interactions $J_{ij}$ fixed (*Figure 5C*); in practice, we defined two sets of additional biases, $\delta \overleftarrow{h}_i$ and $\delta \overrightarrow{h}_i$, corresponding, respectively, to leftward and rightward illuminations. The underlying intuition is that biases encode inputs due to the stimulation, while the interactions between neurons can be considered as fixed over the experimental time scale. This simplified model reproduces the low order statistics of the data under stimulation (*Appendix 2—figure 8A and B*).

The inferred values of the additional biases, averaged over the entire subpopulation (right or left), are shown in *Figure 5D* for both ipsiversive or contraversive stimulation. The results show that light stimulation produces a strong increase of excitability for the ipsilateral neurons and a smaller one for contralateral neurons.

We then simulated the visual stimulation protocol by sampling the Ising model while alternating the model parameters, from $\{h_i + \delta \overrightarrow{h}_i, J_{ij}\}$ to $\{h_i + \delta \overleftarrow{h}_i\}, J_{ij}\}$, and back. The simulated dynamics of the model (*Figure 5E*) qualitatively reproduces the experimental traces of the ARTR activity (*Figure 5B*). In particular, the model captures the stabilizing effect of unilateral visual stimuli, which results in a large activation of the ipsilateral population, which in turn silences the contralateral subcircuit due to the negative $I$ coupling between both. This yields the anticorrelation between the left and right sides clearly visible in both the experimental and simulated traces, and much stronger in the case of spontaneous activity (*Appendix 2—figure 8C–F*).

To better understand the Ising dynamics under visual stimulation, we resort, as previously, to mean-field theory. For asymmetric stimulation, our mean-field model includes, during the periods of stimulation, extra biases $\Delta H_L$ and $\Delta H_R$ over neurons in, respectively, the left and right areas (*Figure 5C*), while the couplings $J$ and $I$ remain unchanged. We show in *Figure 5F* the free-energy $\mathcal{F}$ as a function of $m_L, m_R$ for an example fish. Due to the presence of the extra bias, the landscape is tilted with respect to its no-stimulation counterpart (*Figure 5G*), entailing that the left- or right-active states are much more likely, and the barrier separating them from the low–low state is much lower. As a consequence, the time necessary for reaching the high-activity state is considerably reduced with respect to the no-stimulation case (see *Equation 5*). These results agree with the large probability of the high-activity states and the fast rise to reach these states in the Ising traces in *Figure 5E* compared with *Figure 2C*.

## Discussion

Modeling high-dimensional data, such as extensive neural recordings, imposes a trade-off between accuracy and interpretability. Although highly sophisticated machine-learning methods may offer quantitative and detailed predictions, they might in turn prove inadequate to elucidate fundamental neurobiological mechanisms. Here, we introduced a data-driven network model, whose biologically grounded architecture and relative simplicity make it both quantitatively accurate and amenable to detailed mathematical analysis. We implemented this approach on functional recordings performed at various temperature of a key population of neurons in the zebrafish larvae brain, called ARTR, that drives the orientation of tail bouts and gaze (*Dunn et al., 2016*; *Wolf et al., 2017*; *Ramirez and Aksay, 2021*; *Leyden et al., 2021*).

First, we demonstrate that the persistent time scale of the ARTR endogenous dynamics decreases with the temperature, mirroring the thermal modulation of turn direction persistence in freely swimming behavioral assays. We then demonstrate that our energy-based model not only captures the statistics of the different activity patterns, but also numerically reproduces the endogenous pseudo-oscillatory network dynamics, and their thermal dependence. The inferred Ising model is then analyzed within the so-called mean-field formulation, in which the coupling and bias parameters are replaced by their values averaged over the left and right subpopulations. It yields a two-dimensional representation of the network energy landscape where the preferred states and associated activation barriers can be easily evaluated. We show how this combined data-driven and theoretical approach can be applied to analyze the ARTR response to transient visual stimulation. The latter tilts the energy landscape, strongly favoring some states over others.

## Origin and functional significance of the temperature dependence of the ARTR dynamics

The brains of cold-blooded animals need to operate within the range of temperature that they experience in their natural habitat, for example, 18–33°C for zebrafish (*Gau et al., 2013*). This is a peculiarly stringent requirement since most biophysical processes are dependent on the temperature. In some rare instances, regulation mechanisms might stabilize the circuit dynamics in order to preserve its function, as best exemplified by the pyloric rhythm of the crab whose characteristic phase relationship is maintained over an extended temperature range (*Tang et al., 2010*). Yet in general, an increase in temperature tends to increase the frequency of oscillatory processes (*Robertson and Money, 2012*). The observed acceleration of the ARTR left/right alternation with increasing temperature could thus directly result from temperature-dependent cellular mechanisms. Furthermore, one cannot rule out the possibility that the ARTR dynamics could also be indirectly modulated by temperature via thermal-dependent descending neuromodulatory inputs.

As a result of this thermal modulation of the neuronal dynamics, many cold-blooded animals also exhibit temperature dependence of their behavior (*Long and Fee, 2008*; *Neumeister et al., 2000*; *Stevenson and Josephson, 1990*). Here, we were able to quantitatively relate the two processes (neuronal and motor) by demonstrating that an increase in temperature consistently alters the pattern of spontaneous navigation by increasing the left/right alternation frequency. Interpreting the functional relevance of this modification of the swimming pattern is tricky since many other features of the animal's navigation are concurrently impacted by a change in temperature, such as the bout frequency, turning rate, turn amplitude, etc. Nevertheless, we were able to show in a recent study that this thermal dependence of the swimming kinematic endows the larva with basic thermophobic capacity, thus efficiently protecting them from exposure to the hottest regions of their environment (*Le Goc et al., 2021*).

## Ising model is not trained to reproduce short-term temporal correlations, but is able to predict long-term dynamics

The graphical model we introduced in this work was trained to capture the low-order statistics of snapshots of activity. Because graphical models are blind to the dynamical nature of the population activity, it is generally believed that they cannot reproduce any dynamical feature. Nevertheless, here we demonstrate that our model can quantitatively replicate aspects of the network long-term dynamics such as the slow alternation between the two preferred states. To better understand this apparent paradox, it is necessary to distinguish short and long time scales. At short time scale, defined here as the duration of a time bin (of the order of a few 100 ms), the model cannot capture any meaningful dynamics. The MC algorithm we used to generate activity is an abstract and arbitrary process, and the correlations it produces between successive time bins cannot reproduce the ones in the recording data. Capturing the short-term dynamics would require a biologically grounded model of the cell–cell interactions, or, at the very least, to introduce parameters capturing the experimental temporal correlations over this short time scale (*Marre et al., 2009*; *Mézard and Sakellariou, 2011*).

Yet, the inability of the Ising model to reproduce short time dynamical correlations does not hinder its capacity to predict long-time behavior. The separation between individual neuronal processes (taking place over time scales smaller than 100 ms) and network-scale activity modulation, which happens on time scales ranging from 1 to 20 s, is here essential. The weak dependence of macroscopic processes on microscopic details is in fact well known in many fields outside neuroscience. A classic example is provided by chemical reactions, whose kinetics are often controlled by a slow step due to the formation of the activated complex and to the crossing of the associated energy barrier $\Delta E$, requiring a time proportional to $e^{\Delta E/(kT)}$. All fast processes, whose modeling can be very complex, contribute an effective microscopic time scale $\tau$ in Arrhenius' expression for the reaction time (see *Equation 5*). In this respect, what really matters to predict long time dynamical properties is a good estimate of $\Delta E$ or, equivalently, of the effective energy landscape felt by the system. This is precisely what the Ising model is capable of doing. This explains why, even if temporal information are not explicitly included in the training process, our model may still be endowed with a predictive power over the long-term network dynamics.

## Energy-landscape-based mechanism for persistence

In a preceding article (*Wolf et al., 2017*), we developed a mathematical model of the ARTR in which the left and right ARTR population were represented by a single unit. To account for the ARTR persistent dynamics, an intrinsic adaptation time scale had to be introduced in an ad hoc fashion. While the mean-field version of the inferred Ising model shows some formal mathematical similarity with this two-unit model, it differs in a fundamental aspect. Here, the slow dynamics reflects the itinerant exploration of a two-dimensional energy landscape (*Figure 4C*), for which the barriers separating metastable states scale linearly with the system size. The time to cross these barriers in turn grows exponentially with the system size, as prescribed by Arrhenius law, and can be orders of magnitude larger than any single-neuron relaxation time. Persistence is therefore an emerging property of the neural network.

## Mean-field approximation and beyond

The mean-field approach, through a drastic simplification of the Ising model, allows us to unveil the fundamental network features controlling its coarse-grained dynamics. Within this approximation, the distributions of couplings and of biases are replaced by their average values. The heterogeneities characterizing the Ising model parameters (*Appendix 2—figure 4* and *Appendix 2—figure 5*), and ignored in the mean-field approach, may, however, play an important role in the network dynamics.

In the Ising model, the ipsilateral couplings are found to be broadly distributed such as to possess both negative and positive values. This leads to the presence of so-called frustrated loops, that is, chains of neurons along which the product of the pairwise couplings is negative. The states of activities of the neurons along such loops cannot be set in a way that satisfies all the excitatory and inhibitory connections, hence giving rise to dynamical instabilities in the states of the neurons. The absence of frustrated loops in the network (*Figure 4A*) stabilizes and boosts the activity, an artifact we had to correct for in our analytical treatment by introducing an effective number of neurons $K$, much smaller than the total numbers of neurons $N$ s. Neglecting the variability of the contralateral couplings also constitutes a drastic approximation of the mean-field approach. This is all the more true that the average contralateral coupling $I$ happens to be small compared to its standard deviation.

Couplings are not only broadly distributed but also spatially organized. Ipsilateral couplings $J_{ij}$ decay with the distance between neurons $i$ and $j$ (*Appendix 2—figure 4F*). Similarly, contralateral couplings show strong correlations for short distances between the contralateral neurons (*Appendix 2—figure 4G*). The existence of a local spatial organization in the couplings is not unheard of in computational neuroscience and can have important functional consequences. It is, for instance, at the basis of ring-like attractor models and their extensions to two or three dimensions (*Tsodyks and Sejnowski, 1995*). Combined with the presence of variable biases $h_i$, short-range interactions can lead to complex propagation phenomena, intensively studied in statistical physics in the context of the Random Field Ising Model. (*Schneider and Pytte, 1977*; *Kaufman et al., 1986*). As the most excitable neurons (with the largest biases) fire, they excite their neighbors, who in turn become active, triggering the activation of other neurons in their neighborhood. Such an avalanche mechanism could explain the fast rise of activity in the left or right region, from low- to high-activity state.

## Interpretation of the functional connectivity

The inferred functional couplings $J_{ij}$'s are not expected to directly reflect the corresponding structural (synaptic) connectivity. However, their spatial distribution appears to be in line with the known ARTR organization (*Dunn et al., 2016*; *Kinkhabwala et al., 2011*) characterized by large positive (excitatory) interactions within the left and right population, and by the presence of negative (inhibitory) contralateral interactions. Although the contralateral couplings are found to be, on average, almost null, compared to the ipsilateral excitatory counterparts, they drive a subtle interplay between the left and right regions of the ARTR.

Our neural recordings demonstrate a systematic modulation of the ARTR dynamics with the water temperature, in quantitative agreement with the thermal dependence of the exploratory behavior in freely swimming assays. The model correctly captures this thermal modulation of the ARTR activity, and in particular the decay of the persistence time with the temperature. This owes to a progressive change in the values of both the couplings and the biases, which together deform the energy landscape and modulate the energy barriers between metastable states. The fact that the inferred

functional connectivity between neurons does not display simple temperature dependence is not unexpected as different membrane currents can have different temperature dependence (*Partridge and Connor, 1978*).

In addition, as shown in *Appendix 2—table 2*, the inferred parameters largely vary across datasets. This variability is partially due to the difficulty to separately infer the interactions $J_{ij}$ and the biases $h_i$, a phenomenon not specific to graphical model but also found with other neural, for example, Integrate-and-Fire network models (*Monasson and Cocco, 2011*). This issue can be easily understood within mean-field theory. For simplicity, let us neglect the weak contralateral coupling $I$. The mean activity $m$ of a neuron then depends on the total 'input' $Jm + H$ it receives, which is the sum of the bias $H$ and of the mean ipsilateral activity $m$, weighted by the recurrent coupling $J$. Hence, the combination $Jm + H$ is more robustly inferred than $H$ and $J$ taken separately (*Appendix 2—figure 7E*).

The capacity to quantitatively capture subtle differences in the spontaneous activity induced by external cues is an important asset of our model. Recent studies have shown that spontaneous behavior in zebrafish larvae is not time-invariant but exhibits transitions between different regimes, lasting over minutes and associated with specific brain states. These transitions can have no apparent cause (*Le Goc et al., 2021*) or be induced by external (e.g., stimuli; *Andalman et al., 2019*) or internal cues (e.g., hunger states; *Marques et al., 2019*). Although they engage brain-wide changes in the pattern of spontaneous neural dynamics, they are often triggered by the activation of neuromodulatory centers such as the habenula-dorsal raphe nucleus circuit (*Corradi and Filosa, 2021*). Training Ising models in various conditions may help decipher how such neuromodulation impacts the network functional couplings leading to distinct dynamical regimes of spontaneous activity.

## Data-driven modeling and metastability

With its slow alternating activity and relatively simple architecture, the ARTR offers an ideally suited circuit to test the capacity of Ising models to capture network-driven dynamics. The possibility to experimentally modulate the ARTR persistence time scale further enabled us to evaluate the model ability to quantitatively represent this slow process. The ARTR is part of a widely distributed hindbrain network that controls the eye horizontal saccadic movements, and which includes several other neuronal populations whose activity is tuned to the eye velocity or position (*Joshua and Lisberger, 2015*; *Wolf et al., 2017*). A possible extension of the model would consist in incorporating these nuclei in order to obtain a more complete representation of the oculomotor circuit. Beyond this particular functional network, a similar data-driven approach could be implemented to capture the slow concerted dynamics that characterize numerous neural assemblies in the zebrafish brain (*van der Plas et al., 2021*).

The importance of metastable states in cortical activity in mammals has been emphasized in previous studies as a possible basis for sequence-based computation (*Harvey et al., 2012*; *Brinkman et al., 2022*). Our model suggests that these metastable states are shaped by the connectivity of the network and are naturally explored during ongoing spontaneous activity. In this respect, the modification of the landscape resulting from visual stimulation, leading to a sharp decrease in the barrier separating the states, is reminiscent of the acceleration of sensory coding reported in *Mazzucato et al., 2019*. Our principled data-driven modeling could be useful to assess the generality of such metastable-state-based computations and of their modulation by sensory inputs in other organisms.

## Materials and methods

All data and new codes necessary to reproduce the results reported in this work can be accessed at https://gin.g-node.org/Debregeas/ZF_ARTR_thermo and https://github.com/SebastWolf/ZF_ARTR_thermo.

**Key resources table**

| Reagent type (species) or resource | Designation | Source or reference | Identifiers | Additional information |
|---|---|---|---|---|
| Strain, strain background (*Danio rerio*) | Tg(elavl3:H2B-GCaMP6s) | *Vladimirov et al., 2014* | | |
| Strain, strain background (*D. rerio*) | Tg(elavl3:H2B-GCaMP6f) | *Quirin et al., 2016* | | |

*Continued on next page*

*Continued*

| Reagent type (species) or resource | Designation | Source or reference | Identifiers | Additional information |
|---|---|---|---|---|
| Software, algorithm | Blind Sparse Deconvolution | *Tubiana et al., 2020* | BSD | |
| Software, algorithm | Computational Morphometry Toolkit | https://www.nitrc.org/projects/cmtk/ | CMTK | |
| Software, algorithm | Adaptive Cluster Expansion | *Barton and Cocco, 2013* | ACE | |

## Zebrafish lines and maintenance

All animals subjects were zebrafish (*Danio rerio*), aged 5–7 days post-fertilization (dpf). Larvae were reared in Petri dishes in embryo medium (E3) on a 14/10 hr light/dark cycle at 28°C, and were fed powdered nursery food (GM75) every day from 6 dpf.

Calcium imaging experiments were conducted on *nacre* mutants that were expressing either the calcium indicator GCaMP6f (12 fish) or GCaMP6s (1 fish) in the nucleus under the control of the nearly pan-neuronal promoter *Tg(elavl3:H2B-GCaMP6)*. Both lines were provided by Misha Ahrens and published in *Vladimirov et al., 2014* (H2B-GCaMP6s) and *Quirin et al., 2016* (H2B-GCaMP6f).

All experiments were approved by the Le Comité d'Éthique pour l'Expérimentation Animale Charles Darwin (02601.01).

## Behavioral assays

The behavioral experiments and preprocessing have been described in detail elsewhere (*Le Goc et al., 2021*). Shortly, it consists of a metallic pool regulated in temperature with two Peltier elements, recorded in uniform white light from above at 25 Hz. A batch of 10 animals experienced 30 min in water at either 18, 22, 26, 30, or 33°C (10 batches of 10 fish, involving 170 different individuals, were used). Movies were tracked with FastTrack (*Gallois and Candelier, 2021*), and MATLAB (The MathWorks) was used to detect discrete swim bouts from which the differences of orientation between two consecutive events are computed, referred to as turn or reorientation angles $\delta\theta$.

Turn angles distributions could be fitted as the sum of two distributions (Gaussian and Gamma), whose intersection was used to define an angular threshold to categorize events into forward (F), left turn (L), or right turn (R, *Figure 1E*). This threshold was found to be close to 10° for all tested temperatures.

Then we ternarized $\delta\theta$ values, based on F, L, or R classification (*Figure 1F*), and computed the power spectrum of the binary signals defined from symbols L and R only, with the periodogram MATLAB function and averaged by temperature (*Figure 1G*). The outcome was fitted to the Lorentzian expression corresponding to a memory-less equiprobable two-state process (*Odde and Buettner, 1998*):

$$S(f) \propto \frac{2k_{flip}}{4k_{flip}^2 + (2\pi f)^2},$$

(6)

where $k_{flip}$ is the rate of transition from one state to another. The inverse of the fitted flipping rate $k_{flip}$ represents the typical time spent in the same orientational state, that is, the typical time taken to switch turning direction.

## Light-sheet functional imaging of spontaneous activity

Volumetric functional recordings were carried out using custom-made one-photon light-sheet microscopes whose optical characteristics have been detailed elsewhere (*Panier et al., 2013*). Larvae were mounted in a 1 mm diameter cylinder of low melting point agarose at 2% concentration.

Imaged volume corresponded to 122 ± 46 µm in thickness, split into 16 ± 4 slices (mean ± SD). Recordings were of length 1392 ± 256 s with a brain volume imaging frequency of 6 ± 2 Hz (mean ± SD).

Image preprocessing, neurons segmentation, and calcium transient ($\Delta F/F$) extraction were performed offline using MATLAB, according to the workflow previously reported (*Panier et al., 2013*; *Wolf et al., 2017*; *Migault et al., 2018*).

A Peltier module is attached to the lower part of the pool (made of tin) with thermal tape (3M). A type T thermocouple (Omega) is placed near the fish head (<5 mm) to record the fish surrounding temperature. The signal from a thermocouple amplifier (Adafruit) is used in a PID loop implemented on an Arduino board, which mitigate the Peltier power to achieve the predefined temperature target, stable at ±0.5°C. The temperature regulation software and electronics design are available on Gitlab under a GNU GPLv3 license (https://gitlab.com/GuillaumeLeGoc/arduino-temperature-control copy archived at *Le Goc, 2022*).

The ARTR neurons were selected using a method described elsewhere (*Wolf et al., 2017*). First, a group of neurons was manually selected on a given slice based on a morphological criterion such that the ARTR structure (ipsilateral correlations and contralateral anticorrelation) is revealed. Then, neurons showing Pearson's correlation (anti-correlation) higher than 0.2 (less than –0.15, respectively) are selected, manually filtering them on a morphological criterion. Those neurons are then added to the previous ones, whose signals are used to find neurons from the next slice and so on until all slices are treated.

For fish that were recorded at different temperatures, to ensure that the same neurons are selected, we used the Computational Morphometry Toolkit (CMTK, https://www.nitrc.org/projects/cmtk/) to align following recordings onto the first one corresponding to the same individual. Resulting transformations are then applied to convert neurons coordinates in a consistent manner through all recordings involving the same fish.

## Visually driven recordings

Volumetric functional recordings under visual stimulation were carried using our two-photon lightsheet microscope described in *Wolf et al., 2015*. The stimulation protocol was previously explained in *Wolf et al., 2017*: two LEDs were positioned symmetrically outside of the chamber at 45° and 4.5 cm from the fish eyes, delivering a visual intensity of 20 μW/cm$^2$. We alternately illuminated 17 times each eye for 10 s, 15 s, 20 s, 25 s, and 30 s while performing two-photon light-sheet brain-wide functional imaging. Synchronization between the microscope and the stimulation set-up was done using a D/A card (NI USB-6259 NCS, National Instruments) and a LabVIEW program. Brain volume image frequency was of 1 Hz on the six recorded fish. Recordings last for 4500 s, 856 s of which is spontaneous activity. We extracted the ARTR neurons following the same procedure described above, yielding 89 ± 54 neurons (mean ± SD).

## Time constants definitions

For the flipping rates (*Figure 1D*), we defined the time-dependent signed activity of the ARTR (*Figure 1B*) through

$$\sigma(t) = \text{sign}\big(m_L(t) - m_R(t)\big) , \tag{7}$$

where $m_{L,R}(t) = \frac{1}{N_{L,R}} \sum_{i \in L,R} s_i(t)$ are the average activities in the L, R regions. A power spectrum density is estimated for each signal with the Thomson's multitaper method through the pmtm MATLAB function (time-halfbandwidth product set to 4). The power spectrum densities were then fitted with a Lorentzian spectrum see *Equation 6* and *Figure 1G*.

ARTR left and right persistence times (*Figure 3B*) are defined as the time $m_L$ and $m_R$ signals spend consecutively above an arbitrary threshold set at 0.1. Left and right signals are treated altogether. Changing the threshold does induce a global offset but does not change the observed effect of temperature, the relation with $m_L$ and $m_R$ mean signals, nor the relation with the persistence times of the synthetic signals. The persistence times of the synthetic signals, generated with the Ising models, are computed using the same procedure: we compute the time $m_L$ and $m_R$ synthetic signals spend consecutively above an arbitrary threshold set at 0.1, we then normalize these durations by the corresponding experimental frame rate in order to compare the different recordings (*Figure 3B*). For the mean-field simulated dynamics of the left and right activities, we also follow the same strategy in order to compute the persistence times displayed in *Figure 4F*.

### Inference of Ising model from neural activity

#### From spontaneous activity to spiking data, to biases and connectivity

For each recording (animal and/or temperature), approximate spike trains were inferred from the fluorescence activity signal using the Blind Sparse Deconvolution algorithm (*Tubiana et al., 2020*). This algorithm features automatic (fully unsupervised) estimation of the hyperparameters, such as spike amplitude, noise level, and rise and decay time constants, but also an automatic thresholding for binarizing spikes such as to maximize the precision-recall performance. The binarized activity of the $N$ recorded neurons was then described for each time bin $t$, into a $N$-bit binary configuration $\mathbf{s}_t$, with, $s_i(t) = 1$ if neuron $i$ is active in bin $t$, 0 otherwise.

The functional connectivity matrix $J_{ij}$ and the biases $h_i$ defining the Ising probability distribution over neural configurations (see *Equation 1*) were determined such that the pairwise correlations and average activities computed from the model match their experimental counterparts. In practice, we approximately solved this hard inverse problem using the Adaptative Cluster Expansion and the MC learning algorithms described in *Cocco and Monasson, 2011* and in *Barton and Cocco, 2013*. The full code of the algorithms can be downloaded from the GitHub repository: https://github.com/john-barton/ACE/ ( *Barton, 2019*).

#### Monte Carlo sampling

In order to generate synthetic activity, we resorted to Gibbs sampling, a class of Monte Carlo Markov Chain method, also known as Glauber dynamics. At each time step $k$, a neuron, say, $i$, is picked up uniformly at random, and the value of its activity is updated from $s_i^k$ to $s_i^{k+1} = 0, 1$ according to the probability

$$P\left(s_i^{k+1} \mid s_{j \neq i}^k\right) = \frac{\exp\left(s_i^{k+1}\left(h_i + \sum_j J_{ij} s_j^k\right)\right)}{1 + \exp\left(h_i + \sum_j J_{ij} s_j^k\right)} \tag{8}$$

which depends on the current activities of the other neurons. As this updating fulfills detailed balance, the probability distribution of $\mathbf{s}^k$ eventually converges to $P$ in *Equation 1*. A Monte Carlo round is defined as the number of Monte Carlo steps divided by the total number of neurons, $N$. The code used can be accessed from the link provided at the beginning of the 'Materials and methods' section.

#### Cross-validation and independent model

We cross-validated the Ising models (see *Appendix 2—figure 3*) dividing the datasets in two parts: for each experiment, 75% of each dataset is used as a training set and the remaining 25% is used as a test set. Each training set is used to infer an Ising model. We then compare the mean activity and covariance of the test set with the one computed from the simulated data generated by the models (*Appendix 2—figure 3A and B*). We also show the relative variation of the models' log likelihood computed on the training data and the test data (*Appendix 2—figure 3C*). In addition, as a null hypothesis, we decided to compare the Ising models fitted on the data with the independent model. The independent model depends on the mean activities $\langle s_i \rangle_{\mathrm{data}}$ only and reads

$$P(\mathbf{s}) = \frac{1}{Z} \exp\left(\sum_i h_i s_i\right) , \tag{9}$$

We demonstrate in *Appendix 2—figure 3E–F* the inefficiency of the independent models, comparing the mean activity and covariance of the test set with the one computed from the simulated data generated by the independent models. We also show the relative variation, between the Ising and the independent models, of the log likelihood computed on the training data and the test data (*Appendix 2—figure 3G*).

#### Real data and models comparison

To quantify the quality of the log-probability landscapes reproduction by the Ising models (*Figure 3A*), we used the Kullback–Leibler divergence between (1) a dataset $i$ and the synthetic signals generated with the model trained on that dataset $i$ (green) and (2) the dataset $i$ with synthetic signals generated with every other models (red). With $c_i$ the count in the two-dimensional bin $i$ ($10 \times 10$ bins used) and

$\alpha$ a pseudocount (set to 1), the probability in bin $i$ is defined as $P_i = \frac{c_i + \alpha}{\sum_j (c_j + \alpha)}$. The Kullback–Leibler divergence between a data/model pair is then defined as

$$D_{KL} = \sum_i P_{data,i} \log_{10} \left( \frac{P_{data,i}}{P_{model,i}} \right) \ .$$ (10)

We follow the exact same procedure in order to compare the independent model and their corresponding datasets (**Figure 3A** in blue). In this case, we use synthetic signals generated with the independent model to define $P_{model,i}$.

## Inference of additional biases from visually driven activity recordings

For the visually driven activity recordings, we infer the additional biases $\delta \overleftarrow{h}_i$ from the recordings of the ARTR activity (**Figure 5D**) during, for example, the leftward light stimulations as follows. Let $\overleftarrow{B}$ the number of time bins $t = 1, 2, ..., \overleftarrow{B}$ in the recording, and $\mathbf{s}_t$ the corresponding binarized activity configurations. We define, for each neuron $i$,

$$\rho_i(\delta h) = \sum_{t=1}^{\overleftarrow{B}} \frac{\exp \left( h_i + \sum_j J_{ij} s_j(t) + \delta h \right)}{1 + \exp \left( h_i + \sum_j J_{ij} s_j(t) + \delta h \right)} \ .$$ (11)

$\rho_i(\delta h)$ represents the mean activity of neuron $i$, when subject to a global bias summing $h_i$, the other neurons activities $s_j(t)$ weighted by the couplings $J_{ij}$, and an additional bias $\delta h$, averaged over all the frames $t$ corresponding to left-sided light stimulation. It is a monotonously increasing function of $\delta h$, which matches the experimental average activity $\frac{1}{\overleftarrow{B}} \sum_{t=1}^{\overleftarrow{B}} s_i(t)$ for a unique value of its argument. This value defines $\delta \overleftarrow{h}_i$. The same procedure was followed to infer the additional biases $\delta \overrightarrow{h}_i$ associated to rightward visual stimulations.

## Acknowledgements

SW acknowledges support by a fellowship from the Fondation pour la Recherche Médicale (SPF 201809007064), and GLG by the Systems Biology network of Sorbonne Université. We thank the IBPS fish facility staff for the fish maintenance. We are grateful to Carounagarane Dore for his contribution to the design of the experimental setups. We thank Misha Ahrens for providing the GCaMP line.

## Additional information

### Funding

| Funder | Grant reference number | Author |
| --- | --- | --- |
| Agence Nationale de la Recherche | Locomat | Rémi Monasson |

The funders had no role in study design, data collection and interpretation, or the decision to submit the work for publication.

### Author contributions

Sebastien Wolf, Data curation, Software, Formal analysis, Validation, Investigation, Visualization, Methodology, Writing - review and editing; Guillaume Le Goc, Data curation, Validation, Investigation, Visualization, Methodology, Writing - review and editing; Georges Debrégeas, Conceptualization, Funding acquisition, Investigation, Methodology, Writing – original draft, Project administration; Simona Cocco, Conceptualization, Supervision, Methodology, Writing – original draft; Rémi Monasson, Conceptualization, Formal analysis, Supervision, Investigation, Methodology, Writing – original draft, Project administration

## Author ORCIDs

Sebastien Wolf http://orcid.org/0000-0001-9394-3291
Guillaume Le Goc http://orcid.org/0000-0002-6946-1142
Georges Debrégeas http://orcid.org/0000-0003-3698-4497
Simona Cocco http://orcid.org/0000-0002-1852-7789
Rémi Monasson http://orcid.org/0000-0002-4459-0204

## Ethics

All experiments were approved by Le Comité d'Éthique pour l'Expérimentation Animale Charles Darwin (02601.01).

## Decision letter and Author response

Decision letter https://doi.org/10.7554/eLife.79541.sa1
Author response https://doi.org/10.7554/eLife.79541.sa2

---

# Additional files

## Supplementary files

• MDAR checklist

## Data availability

All data necessary to reproduce the results reported in this work can be accessed from https://gin.g-node.org/Debregeas/ZF_ARTR_thermo. Codes can be accessed at https://github.com/SebastWolf/ZF_ARTR_thermo copy archived at *Wolf, 2023*.

The following dataset was generated:

| Author(s) | Year | Dataset title | Dataset URL | Database and Identifier |
|---|---|---|---|---|
| Wolf S, Le Goc G, Debregeas G, Cocco S, Monasson R | 2023 | ZF_ARTR_thermo | https://gin.g-node.org/Debregeas/ZF_ARTR_thermo | G-Node GIN, Debregeas/ZF_ARTR_thermo |

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

## Appendix 1

## Mean-field theory for the ARTR activity

### Derivation of the free energy

We consider an Ising model with $N_L$ and $N_R$ neurons in, respectively, the left and right regions. Each neuron activity variable can take two values, $i = 0, 1$, corresponding to silent and active states (within a time window). The 'energy' of the system reads

$$E(s_1, \ldots, s_{N_L}, s_{N_L+1}, \ldots, s_{N_L+N_R}) = -\tilde{H}_L \sum_{i=1}^{N_L} s_i - \tilde{H}_R \sum_{i=N_L+1}^{N_L+N_R} s_i - \frac{1}{2} \sum_{i \neq j} \tilde{J}_{ij} s_i s_j \,, \qquad (12)$$

where $\tilde{H}_L, \tilde{H}_R$ are biases acting on the neurons, and the coupling matrix is defined through

$$\tilde{J}_{ij} = \begin{cases} \tilde{J}_L & \text{if} & 1 \leq i, j \leq N_L, \\ \tilde{J}_R & \text{if} & N_L + 1 \leq i, j \leq N_L + N_R, \\ \tilde{I} & \text{otherwise} \end{cases} \qquad (13)$$

We now introduce the left and right average activities:

$$m_L = \frac{1}{N_L} \sum_{i=1}^{N_L} s_i \,, \qquad m_R = \frac{1}{N_R} \sum_{i=N_L+1}^{N_L+N_R} s_i \,. \qquad (14)$$

The energy $E$ of a neural activity configuration in **Equation 12** can be expressed in terms of these average activities:

$$\begin{aligned} E(m_L, m_R) &= -N_L \left( \tilde{H}_L - \frac{\tilde{J}_L}{2} \right) m_L - N_R \left( \tilde{H}_R - \frac{\tilde{J}_R}{2} \right) m_R \\ &\quad - \frac{(N_L)^2}{2} \tilde{J}_L m_L^2 - \frac{(N_R)^2}{2} \tilde{J}_R m_R^2 - \tilde{I} N_L N_R m_L m_R \,. \end{aligned} \qquad (15)$$

We may now compute the partition function normalizing the probability of configurations,

$$Z = \sum_{\{s_i=0,1\}} e^{-E(s_1,\ldots,s_{N_L+N_R})} = \sum_{m_L,m_R} \mathcal{M}_L(m_L) \, \mathcal{M}_R(m_R) \, e^{-E(m_L,m_R)} \,, \qquad (16)$$

where the sums run over fractional values of the average left and right activities, from 0 to 1 with steps equal to, respectively, $2/N_L$ and $2/N_R$, and the multiplicities $\mathcal{M}_L$ and $\mathcal{M}_R$ measure the numbers of neural configurations with prescribed average activities. We approximate these multiplicities with the standard entropy-based expressions, which are exact in the limit of large sizes $K_L, K_R$:

$$\mathcal{M}_L(m_L) \simeq e^{N_L \, S(m_L)} \,, \qquad \mathcal{M}_R(m_R) \simeq e^{N_R \, S(m_R)} \,, \qquad (17)$$

where

$$S(m) = -m \ln m - (1 - m) \ln(1 - m) \qquad (18)$$

is the entropy of a $0 - 1$ variable with mean $m$. As a consequence, the activity-dependent free energy is given by

$$\begin{aligned} \mathcal{F}(m_L, m_R) &= E(m_L, m_R) - N_L S(m_L) - N_R S(m_R) \\ &= -\frac{N_L J_L}{2} m_L^2 - \frac{N_R J_R}{2} m_R^2 - I \sqrt{N_L N_R} \, m_L m_R - N_L H_L m_L - N_R H_R m_R \\ &\quad + N_L \left( m_L \ln m_L + (1 - m_L) \ln(1 - m_L) \right) + N_R \left( m_R \ln m_R + (1 - m_R) \ln(1 - m_R) \right) \end{aligned} \qquad (19)$$

where the bias and coupling parameters are, respectively, $H_L = \tilde{H}_L - \frac{\tilde{J}_L}{2}, H_R = \tilde{H}_R - \frac{\tilde{J}_R}{2}, J_L = N_L \tilde{J}_L, J_R = N_R \tilde{J}_R, I = \sqrt{N_L N_R} \tilde{I}$.

The sizes $N_L, N_R$ enter formula (19) for the free energy in two ways:

- Implicitly, through the biases $H_L, H_R$ and the couplings $J_L, J_R, I$. These parameters are equal to, respectively, the average bias and the total ipsilateral and contralateral couplings acting on each neuron in the $L$ and $R$ regions. They are effective parameters defining the mean-field theory.

- Explicitly, as multiplicative factors to the free energy contributions coming from the left and right regions. The sizes then merely act as effective inverse 'temperatures,' in the Boltzmann factor $e^{-F(m_L, m_R)}$ associated to the probability of the $L, R$ activities.

Mean-field theory generally overestimates the collective effects of interactions; a well-known illustration of this artifact is the prediction of the existence of a phase transition in the unidimensional ferromagnetic Ising model with short-range interactions, while such a transition is rigorously known not to take place (*Ma, 1985*). We expect these effects to be strong here due to the wide distribution of inferred Ising couplings (*Appendix 2—figure 4A*). Many pairs of neurons carry close to zero couplings, and the interaction neighborhood of a neuron is effectively much smaller than $N_L$ and $N_R$. To compensate for the overestimation of interaction effects, we thus propose to keep *Equation 19* for the free energy, but with effective sizes $K_L, K_R$ replacing the numbers $N_L, N_R$ of recorded neurons (see *Equation 2*), leading to the expression of the free energy:

$$
\begin{aligned}
\mathcal{F}\left(m_L, m_R\right) &= -\frac{K_L J_L}{2} m_R^2 - \frac{K_R J_R}{2} m_R^2 - I\sqrt{K_L K_R} m_L m_R - K_L H_L m_L - K_R H_R m_R \\
&\quad + K_L \left(m_L \, 1n \, m_L + (1 - m_L) \, 1n(1 - m_L)\right) + K_R \left(m_R \, 1n \, m_R + (1 - m_R) \, 1n(1 - m_R)\right)
\end{aligned}
\tag{20}
$$

These effective sizes $K_L, K_R$ are expected to be smaller than $N_L, N_R$. Their values are fixed through the comparison of the Langevin dynamical traces with the traces coming from the data; see below.

## Langevin dynamical equations

The dynamical Langevin equations read

$$
\tau \frac{dm_L}{dt} = K_L\left(J_L m_L + H_L\right) + I\sqrt{K_L K_R} m_R - K_L \log\left(\frac{m_L}{1 - m_L}\right) + \epsilon_L(t) \,,
\tag{21}
$$

$$
\tau \frac{dm_R}{dt} = K_R\left(J_R m_R + H_R\right) + I\sqrt{K_L K_R} m_L - K_R \log\left(\frac{m_R}{1 - m_R}\right) + \epsilon_R(t) \,,
\tag{22}
$$

where $\epsilon_L, \epsilon_R$ denote white-noise processes; see main text.

## Fit of the effective sizes $K_L$ and $K_R$

The effective sizes $K_L = N_L/A$ and $K_R = N_R/A$ were fitted generating Langevin trajectories of the activities $(m_L, m_R)$ for a large set of values of $A$ (i.e., $K_L$ and $K_R$), and with fixed parameters $(H_L, H_R, J_L, J_R, \tau)$. For each value of $K_L$ and $K_R$, we computed the KL divergence between the experimental and the Langevin distributions of $(m_L, m_R)$ (see *Appendix 2—figure 7A–C*). The effective sizes $K_L$ and $K_R$ are the ones that minimize the value of the KL divergence. For low values of $A$, the KL divergence can be noisy and creates artifacts. To avoid these artifacts, we assume that $A > 2$.

# Appendix 2

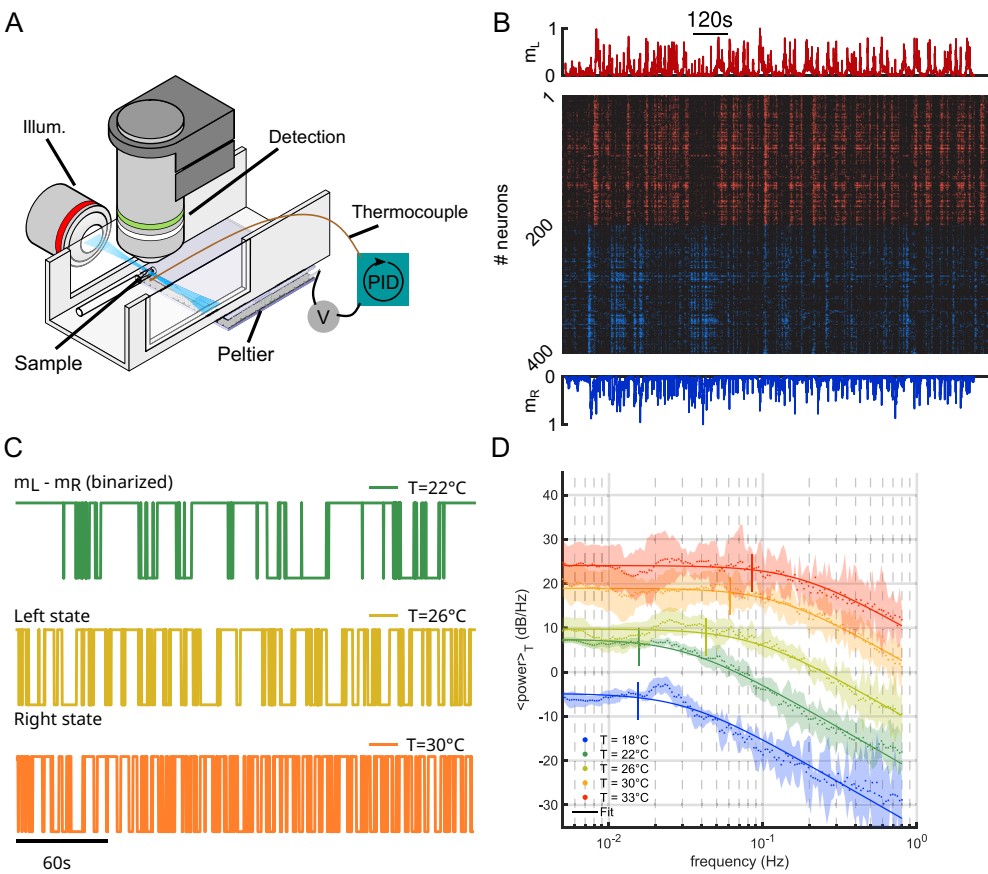

**Appendix 2—figure 1.** Temperature dependence of the anterior rhombencephalic turning region (ARTR) activity. (**A**) Schematic of the experimental setup used to perform brain-wide calcium imaging of a zebrafish larva at controlled water temperature. (**B**) Raster plot of the ARTR spontaneous dynamics showing alternating right/left activation. The top and bottom traces are the ARTR average signal of the left and right subcircuits. (**C**) Example ARTR sign($m_L - m_R$) binarized signals measured at three different temperatures (same larva). (**D**) Averaged power spectrum of the ARTR signals $m_R - m_L$ for the five tested temperatures. Lorentzian fits are shown as solid lines.

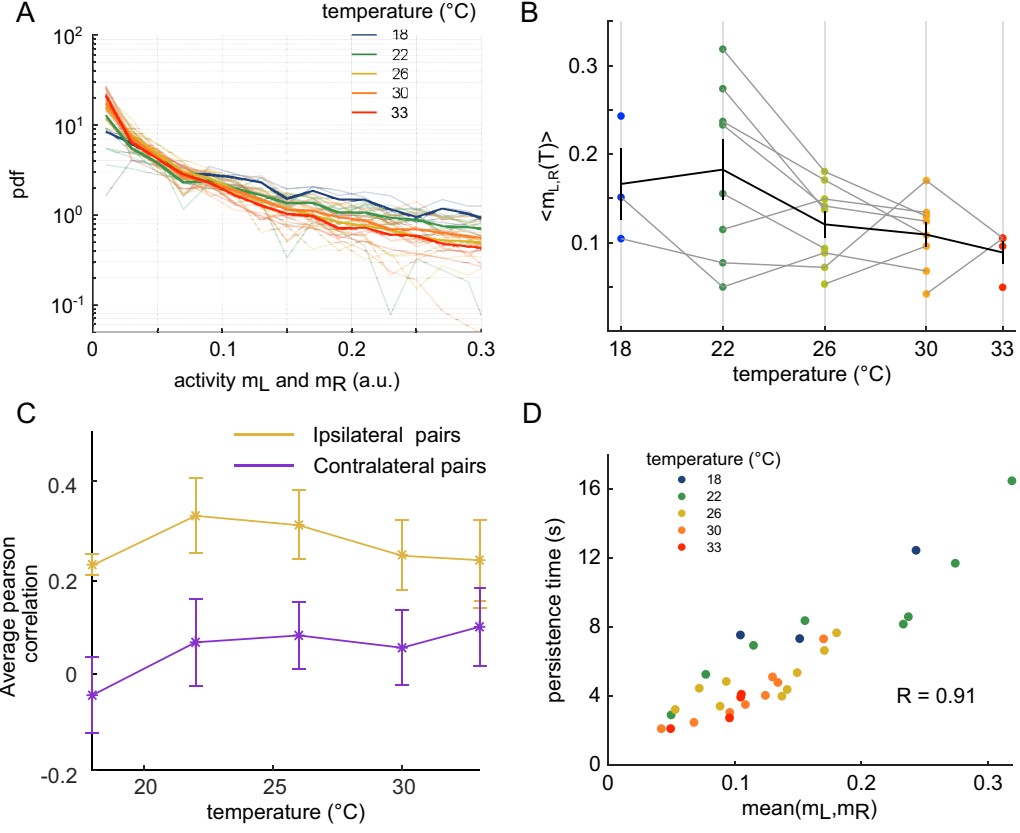

**Appendix 2—figure 2.** Effect of temperature on the anterior rhombencephalic turning region (ARTR) time persistence and activity. (**A**) Pdf of activities of both sides of the ARTR. Color encodes temperature. (**B**) Temperature-averaged mean activity of ARTR left and right neuronal subpopulations. Error bars are standard error of the mean. (**C**) Temperature-averaged Pearson correlation for left/right ispilateral pairs (yellow line) or for contralateral pairs of neurons (purple line). Error bars are standard deviations (32 recordings), n = 13 fish at 5 different water temperatures. (**D**) ARTR persistence time vs. mean activity; note the quasi-linear dependence of these quantities ($R = 0.91$). Each dot is the mean persistence time computed for one fish at one temperature; colors encode temperature.

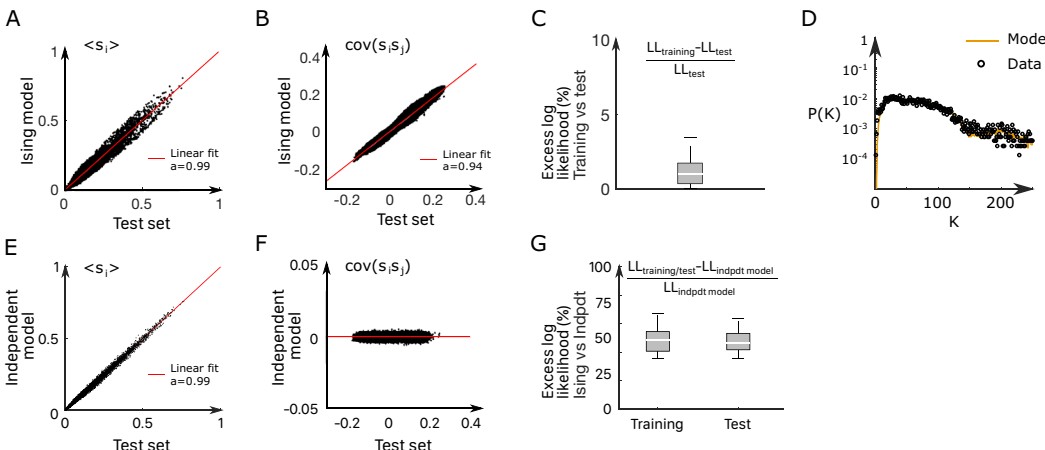

**Appendix 2—figure 3.** Inference of the anterior rhombencephalic turning region (ARTR) Ising model. (**A, B**) Comparison between the mean activities (**A**) and pairwise correlations (**B**) computed from experimental test data and from synthetic (Ising model-generated) data (32 recordings, n = 13 fish). Ising models were trained on a distinct subset of the experimental data. (**C**) Relative variation of the log–likelihoods of the Ising models between training and test data, showing the absence of overfitting. (**D**) Probability that $K$ of the $N$ neurons in the ARTR are simultaneously active in the data (black dots) and in the model (yellow line) configurations. (**E, F**) In order to demonstrate the need for effective connections in our model, we generated synthetic data with independent models of the training dataset. Here, we compare the mean activity (**E**) and the pairwise covariance (**F**) computed on the experimental test dataset and using independent models. (**G**) Excess log likelihood of the Ising models compared to the independent model for training and test data set (see 'Materials and methods').

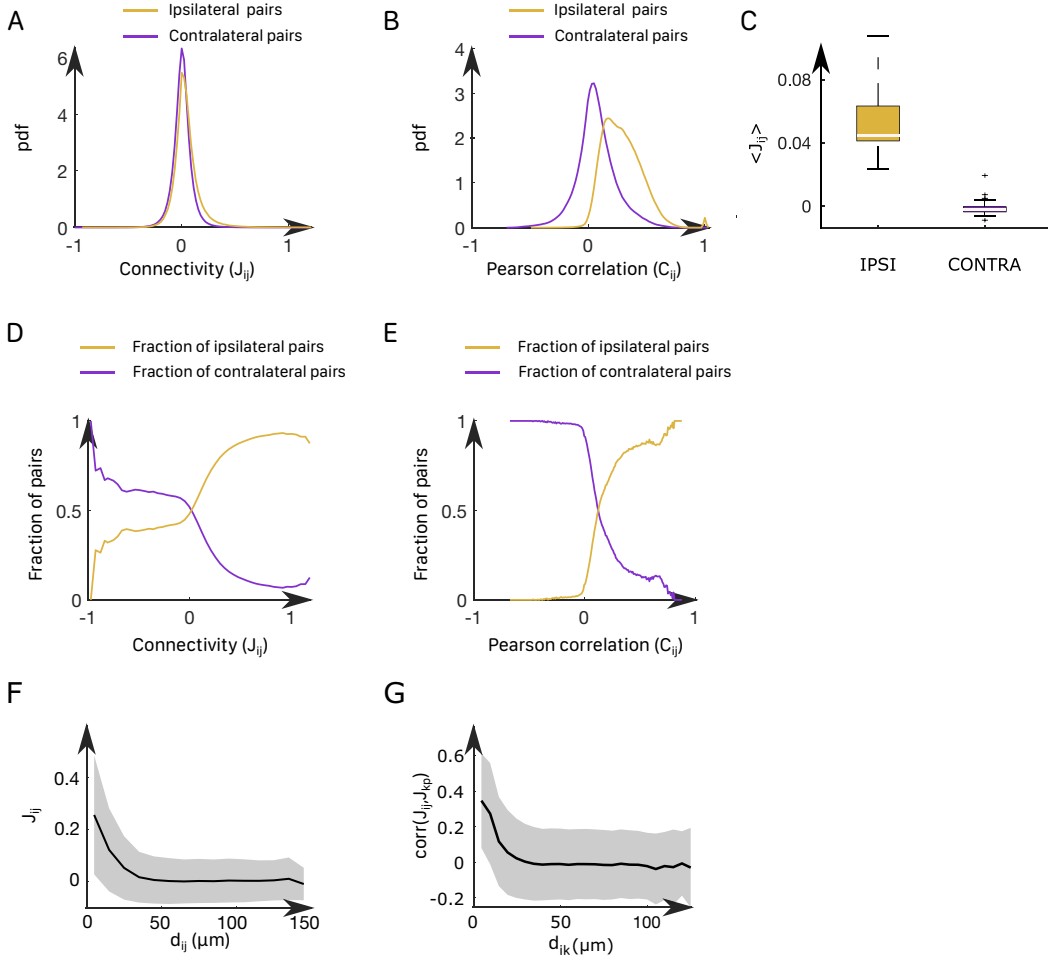

**Appendix 2—figure 4.** Correlation structure within the anterior rhombencephalic turning region (ARTR) and properties of the inferred couplings. (**A**) Probability density function of the functional connectivity for the ipsilateral (gold line) and the contralateral (purple line) couplings. These pdf were obtained by averaging across all animals. (**B**) Probability density function of the functional Pearson correlation for the ipsilateral (gold line) and the contralateral (purple line) couplings. (**C**) Box plot across experiments of the average value of the ipsilateral and contralateral couplings. (**D**) Probability to have an ipsilateral (gold line) or a contralateral (purple line) pair of neuron given its effective connectivity. For a given range of the effective connectivity, we compute the number of ipsilateral and contralateral pairs of neurons. (**E**) Probability to have an ipsilateral (gold line) or a contralateral (purple line) pair of neuron given its Pearson correlation. (**F**) Functional connectivity $J_{ij}$ as a function of the distance between neurons $i, j$. (**G**) Correlation between the couplings $J_{ij}$ and $J_{kp}$, between one neuron $i$ and one neuron $k$ as a function of their distance $d_{ik}$ for every possible pair $(i, k)$.

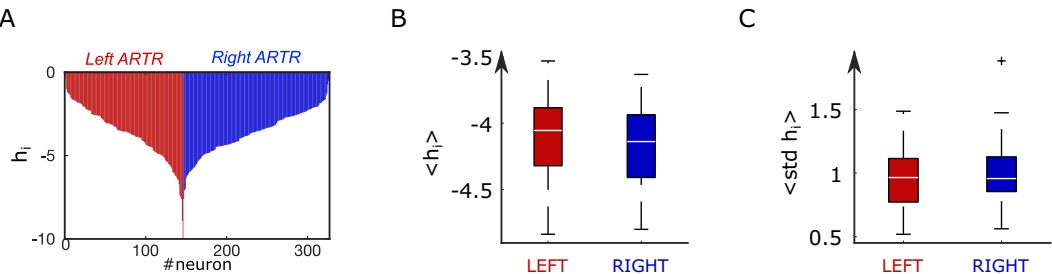

**Appendix 2—figure 5.** Distribution of biases in the inferred anterior rhombencephalic turning region (ARTR) Ising model. (**A**) Bias parameter distribution for an example fish. (**B**) Box plot across experiments of the average value of the biases for the left and right subpopulations of the ARTR. (**C**) Box plot across animals of the standard deviation of the biases for the left and right subpopulations of the ARTR.

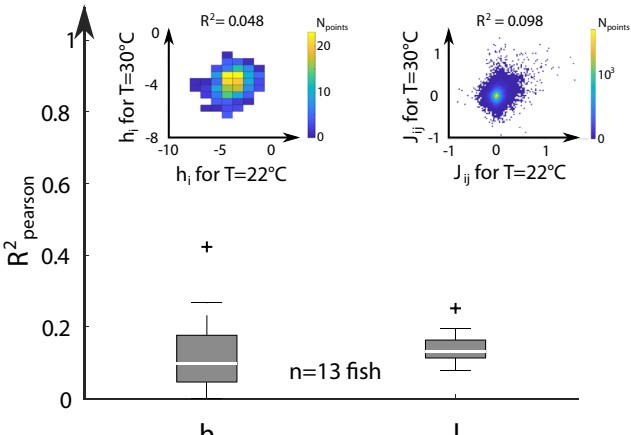

**Appendix 2—figure 6.** Correlation of Ising parameters at different temperatures. For each fish (n = 13), we extract from the scatter plots of the coupling $J_{ij}$ and bias $h_i$ inferred from activity recordings at two different temperatures, the Pearson correlation coefficients $R_{pearson}$. The distribution of $R_{pearson}^2$ values are shown for all fish and pairs of temperature. Inset: Example scatter plots of the inferred biases $h_i$ (left) and effective couplings $J_{ij}$ (right) for the same fish at two different temperature $T = 22$ and $T = 30°C$.

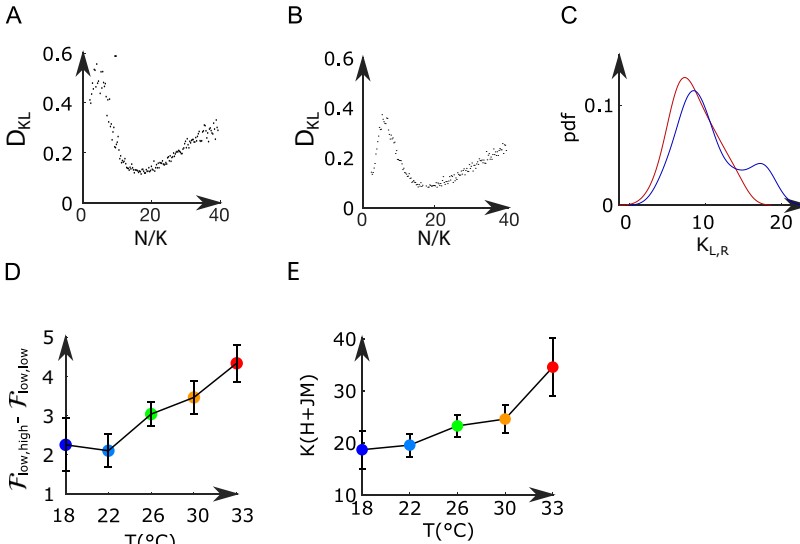

**Appendix 2—figure 7.** Mean-field model of the anterior rhombencephalic turning region (ARTR). (**A, B**) Kullback–Leibler divergence between the experimental and the Langevin distributions as a function of $N/K$, where $N$ is the total number of neurons of the left or right subpopulation, and $K$ is the effective extent of neuronal interaction (see 'Materials and methods') for two datasets. (**C**) Probability density function of $K_R$ (blue line) and $K_L$ (red line) across all recordings. (**D**) Free-energy difference between stationary sates of the landscape as a function of the temperature (32 recordings, n=13 fish). (**E**) Average values (for all experiments and regions) of $K(H + JM)$ as a function of the temperature of the water. Error bars are standard error of the mean.

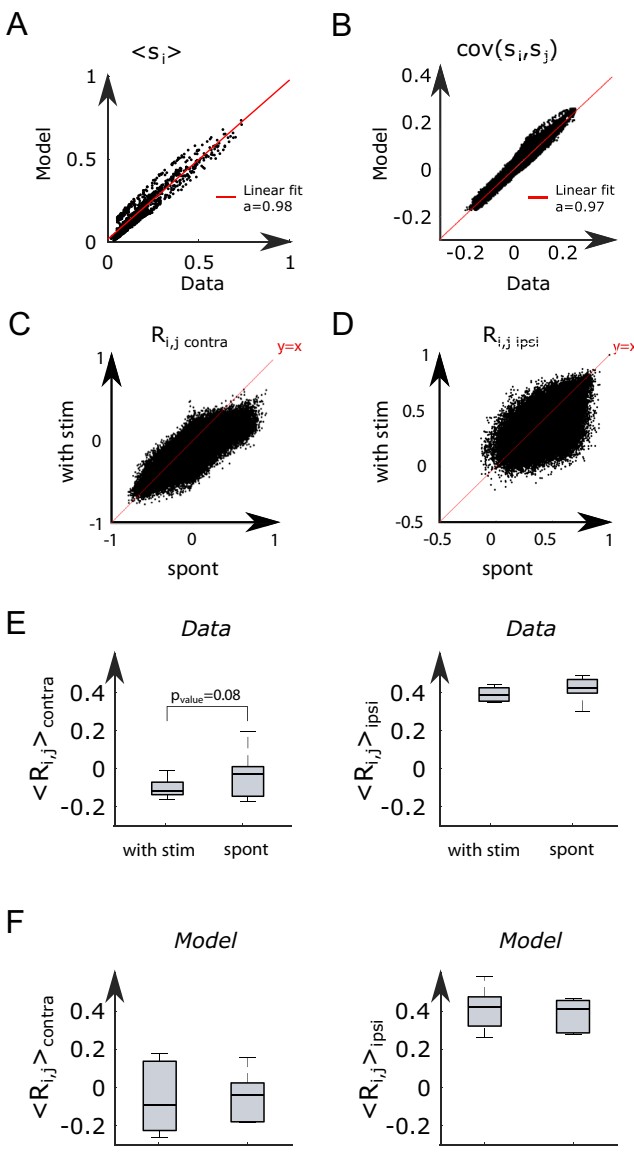

**Appendix 2—figure 8.** A modified Ising model explains visually driven properties of the anterior rhombencephalic turning region (ARTR). (**A, B**) To assess the performance of the model for visually driven experiments, we compare the mean activity (**A**) and the pairwise covariance (**B**) computed on the spontaneous part of the recordings to synthetic data. (**C**) Scatter plot of the correlation between contralateral pairs of neurons under visual stimulation vs. spontaneous activity on $n = 6$ fish. (**D**) Scatter plot of the correlation between ipsilateral pairs of neurons under visual stimulation vs. spontaneous activity. (**E**) Average Pearson correlation in the experimental recordings between contralateral (the p-value of a paired sampled $t$-test is provided) and ipsilateral pairs of cells during stimulated and spontaneous activity ($n = 6$ fish). (**F**) Average Pearson correlation in the simulated activity of the ARTR between contralateral and ipsilateral pairs of cells during stimulated and spontaneous activity ($n = 6$ fish).

**Appendix 2—table 1.** Datasets properties.

| Temperature (°C) | ID | Line | Age (dpf) | $N_L$ | $N_R$ | Acquisition rate (Hz) | Duration (s) |
|---|---|---|---|---|---|---|---|
| 18 | 12 | NucFast | 6 | 146 | 180 | 5 | 1200 |
| 18 | 13 | NucFast | 7 | 37 | 96 | 8 | 1200 |
| 18 | 14 | NucFast | 6 | 179 | 174 | 8 | 1200 |

*Appendix 2—table 1 Continued on next page*

*Appendix 2—table 1 Continued*

| Temperature (°C) | ID | Line | Age (dpf) | $N_L$ | $N_R$ | Acquisition rate (Hz) | Duration (s) |
|---|---|---|---|---|---|---|---|
| 22 | 2 | Nuc slow | 7 | 177 | 212 | 3 | 1106 |
| 22 | 3 | NucFast | 5 | 152 | 85 | 3 | 1812 |
| 22 | 5 | NucFast | 5 | 158 | 123 | 5 | 1500 |
| 22 | 6 | NucFast | 5 | 98 | 134 | 5 | 1500 |
| 22 | 7 | NucFast | 6 | 122 | 221 | 5 | 1500 |
| 22 | 11 | NucFast | 6 | 295 | 320 | 5 | 1200 |
| 22 | 13 | NucFast | 7 | 37 | 96 | 8 | 1200 |
| 22 | 14 | NucFast | 6 | 179 | 174 | 8 | 1200 |
| 26 | 2 | Nuc slow | 7 | 177 | 212 | 3 | 1812 |
| 26 | 3 | NucFast | 5 | 152 | 85 | 3 | 1812 |
| 26 | 4 | NucFast | 5 | 110 | 76 | 3 | 1812 |
| 26 | 5 | NucFast | 5 | 158 | 123 | 5 | 1500 |
| 26 | 6 | NucFast | 5 | 98 | 134 | 5 | 1500 |
| 26 | 7 | NucFast | 6 | 122 | 221 | 5 | 1500 |
| 26 | 11 | NucFast | 6 | 295 | 320 | 5 | 1200 |
| 26 | 13 | NucFast | 7 | 37 | 96 | 8 | 1200 |
| 26 | 14 | NucFast | 6 | 179 | 174 | 8 | 1200 |
| 30 | 2 | Nuc slow | 7 | 177 | 212 | 3 | 1812 |
| 30 | 4 | NucFast | 5 | 110 | 76 | 3 | 1812 |
| 30 | 5 | NucFast | 5 | 158 | 123 | 5 | 1500 |
| 30 | 6 | NucFast | 5 | 98 | 134 | 5 | 1500 |
| 30 | 7 | NucFast | 6 | 122 | 221 | 5 | 1500 |
| 30 | 13 | NucFast | 7 | 37 | 96 | 8 | 1200 |
| 30 | 14 | NucFast | 6 | 179 | 174 | 8 | 1200 |
| 30 | 15 | NucFast | 7 | 202 | 252 | 8 | 1200 |
| 33 | 14 | NucFast | 6 | 179 | 174 | 8 | 1200 |
| 33 | 15 | NucFast | 7 | 202 | 252 | 8 | 1200 |
| 33 | 16 | NucFast | 6 | 127 | 123 | 7 | 1200 |
| 33 | 17 | NucFast | 5 | 62 | 170 | 10 | 1200 |

**Appendix 2—table 2.** Parameters of mean-field models.

| Temperature (°C) | ID | $J_L$ | $J_R$ | $I$ | $H_L$ | $H_R$ | $K_L$ | $K_R$ |
|---|---|---|---|---|---|---|---|---|
| 18 | 12 | 7.06 | 7.23 | –0.6 | –3.66 | –3.63 | 6.51 | 8.03 |
| 18 | 13 | 6.2 | 7.84 | 0.6 | –3.53 | –4.34 | 3.18 | 8.27 |
| 18 | 14 | 7.27 | 7.24 | 0.31 | –3.88 | –3.99 | 11.04 | 10.74 |
| 22 | 2 | 8.2 | 8.28 | 0.12 | –4.24 | –4.23 | 6.65 | 7.96 |
| 22 | 3 | 8.18 | 7.14 | 0.55 | –4.26 | –4.13 | 9.38 | 5.24 |
| 22 | 5 | 7.59 | 7.01 | 0.4 | –4.03 | –3.8 | 5.56 | 4.33 |
| 22 | 6 | 7.13 | 8.69 | 1.1 | –4.49 | –4.64 | 5.21 | 7.12 |

*Appendix 2—table 2 Continued on next page*

*Appendix 2—table 2 Continued*

| Temperature (°C) | ID | $J_L$ | $J_R$ | $I$ | $H_L$ | $H_R$ | $K_L$ | $K_R$ |
|---|---|---|---|---|---|---|---|---|
| 22 | 7 | 7.09 | 7.46 | 0.43 | –3.73 | –3.95 | 6.28 | 11.39 |
| 22 | 11 | 7.82 | 7.59 | –0.1 | –4.07 | –3.91 | 8.28 | 8.98 |
| 22 | 13 | 6.54 | 7.82 | 1.45 | –4.29 | –4.5 | 7.11 | 18.46 |
| 22 | 14 | 7.41 | 8.03 | 0.47 | –4.28 | –4.43 | 10.91 | 10.6 |
| 26 | 2 | 8.37 | 8.22 | –0.49 | –4.47 | –4.31 | 9.72 | 11.64 |
| 26 | 3 | 8.42 | 7.49 | 0.53 | –4.56 | –4.62 | 8.26 | 4.61 |
| 26 | 4 | 8.63 | 6.44 | 0.85 | –4.83 | –4.79 | 10.37 | 7.16 |
| 26 | 5 | 7.29 | 7.59 | 0.48 | –3.92 | –4.14 | 9.08 | 7.06 |
| 26 | 6 | 7.43 | 7.86 | 0.41 | –3.99 | –4.1 | 8.59 | 11.75 |
| 26 | 7 | 7.55 | 7.96 | 0.32 | –4.08 | –4.22 | 4.45 | 8.06 |
| 26 | 11 | 7.27 | 7.45 | 0.37 | –3.89 | –3.92 | 10.31 | 11.18 |
| 26 | 13 | 6.99 | 7.3 | 0.6 | –3.99 | –3.94 | 6.37 | 16.55 |
| 26 | 14 | 7.91 | 7.35 | 0.5 | –4.34 | –4.16 | 11.32 | 11.01 |
| 30 | 2 | 7.54 | 7.96 | –0.12 | –4.54 | –4.56 | 7.02 | 8.41 |
| 30 | 4 | 8.36 | 7.73 | 0.11 | –4.52 | –4.18 | 9.64 | 6.66 |
| 30 | 5 | 6.77 | 6.42 | 0.66 | –3.8 | –3.87 | 9.18 | 7.15 |
| 30 | 6 | 7.35 | 7.38 | 0.45 | –3.91 | –3.97 | 7.53 | 10.3 |
| 30 | 7 | 7.43 | 8.07 | 0.42 | –3.93 | –4.38 | 7.09 | 12.84 |
| 30 | 13 | 6.91 | 7.41 | 0.73 | –4.13 | –4.03 | 5.78 | 15 |
| 30 | 14 | 7.51 | 7.45 | 0.11 | –3.87 | –3.89 | 9.42 | 9.15 |
| 30 | 15 | 8.01 | 8.33 | 0.58 | –4.45 | –4.46 | 13.83 | 17.26 |
| 33 | 14 | 6.74 | 7.02 | 0.76 | –3.8 | –3.97 | 9.32 | 9.06 |
| 33 | 15 | 6.99 | 7.47 | –0.02 | –3.68 | –3.91 | 14.85 | 18.52 |
| 33 | 16 | 7.53 | 8.25 | –0.11 | –4.16 | –4.43 | 14.43 | 13.97 |
| 33 | 17 | 6.66 | 7.36 | 0.45 | –3.69 | –3.89 | 11.92 | 32.69 |

**Appendix 2—table 3.** Parameters of mean-field models.

| ID | $J_L$ | $J_R$ | $I$ | $H_L$ | $H_R$ | $K_L$ | $K_R$ |
|---|---|---|---|---|---|---|---|
| 1 | 7.54 | 7.35 | –0.67 | –3.75 | –3.44 | 5.60 | 3.43 |
| 2 | 7.10 | 7.42 | 0.64 | –3.69 | –4.02 | 7.91 | 12.82 |
| 3 | 7.51 | 7.92 | –0.28 | –3.96 | –4.08 | 4.98 | 3.90 |
| 4 | 8.38 | 6.25 | –0.04 | –3.68 | –3.18 | 13.33 | 4.44 |
| 5 | 8.73 | 8.24 | 0.01 | –4.38 | –4.13 | 6.11 | 6.89 |
| 6 | 7.87 | 7.71 | 0.51 | –4.17 | –4.09 | 16.19 | 15.52 |

