## [Editor Report]

The authors show how high-dimensional neural signals can be reduced to low-dimensional models with variables that can be directly linked to behavior. The reduced model can account for long time scales of persistent activity that arise from transitions between metastable model states. The authors further show that the rate of these transitions is modulated by water temperature according to the classic Arrhenius law, although the results for different temperatures could not yet be unified into a single description based on real external temperature.

---

## [Decision Letter]

**Decision letter after peer review:**

Thank you for submitting your article "Emergence of time persistence in a data-driven neural network model" for consideration by *eLife*. Your article has been reviewed by 2 peer reviewers, and the evaluation has been overseen by a Reviewing Editor and Joshua Gold as the Senior Editor. The reviewers have opted to remain anonymous.

Essential revisions:

(1) Please follow suggestions from reviewing on cross-validating model fitting

(2) Fitting the model across temperature values and discussing the relationship between the physical and model temperatures

(3) Analysis of temporal alignments between changes in the swim direction and the onset of sign change in the difference between the mean population activities of left and right hemispheres

*Reviewer #1 (Recommendations for the authors):*

1. Temporal alignment between neural activity and fish behavior.

Line 101: "Impact of the bath temperature on the orientational persistence in freely swimming larvae".

The authors compare the statistics of their sign(m_L-m_R) to the statistics of swim bouts in a previous paper and show that the two variables are highly correlated. However, this is just an indirect observation since they didn't establish whether changes flips in sign(m_L-m_R) are temporally aligned to changes in the fish swimming direction when analyzed simultaneously. I believe this is an important missing link in the manuscript and I suggest that the authors show that this relationship holds at least in one example – although it'd be ideal to show this at different temperature, it is sufficient to show it for one particular temperature. The dataset in Figure 4 is appropriate for this extra analysis, which should be performed both during spontaneous and stimulated epochs.

Related to this issue: what is the probability of having a change in swim direction, without a flip of sign(m_L-m_R)? And vice versa, what is the probability of having a flip of sign(m_L-m_R) without a change in swim direction?

Also related, what is the distribution of intervals between changes in the swim bout directions? If these behavioral observable is faster than the temporal resolution of the calcium indicator dynamics, this might introduce some statistical biases in the analysis.

2. Cross-validation

Line 143: "Inference of an Ising model from functional recordings of the ARTR." The authors fit an Ising model to the data showing that it captures various observables in the data. However, the model is not cross-validated. The authors should use standard cross-validation procedures to establish the goodness of fit of the various observables on held-out data. In all plots in Figure 2C-F and Appendix 2 Figure 2B-E results should be replaced with a comparison with held-out data not used to fit the model.

Moreover, what is the null hypothesis here? The authors should compare the held-out performance of their model to shuffled models trained on surrogate datasets obtained eg by destroying pairwise correlations.

3. Definition of temperature in the model

It is not explained at all how the water temperature appears in the model. Is it a missing 1/T factor in Equation (1)? This is a central issue the authors should be very clear about. What's a bit confusing is that the authors define "inverse temperature" as the effective size K, but there's no mention of the actual water temperature at all.

Line 234-236: In the Langevin formalism, the diffusion constant D usually appears in the noise autocorrelation function = 2*D*δ(t-t'). This is usually taken to be equal to the temperature (inverse β) in the partition function (which is missing from its definition!). How are the statistical mechanical temperature and the actual water temperature entering here? This is very confusing, I strongly urge the authors to clarify this point.

*Reviewer #2 (Recommendations for the authors):*

This is a beautiful study that combines physics with biology, treating a biological network as an inorganic system, by minimizing its free energy and maximizing entropy. It answers a lot of questions and hints at a valuable wider applicability of this approach. At the same time I am somewhat confused about the approach the authors have taken to fitting the models, explained below, and it would be good to have a conversation about that and possibly adjust some details of the approach, or to clarify. With appropriate revisions or clarifications this work would be a strong candidate for the journal.

Comments, questions, suggestions, and concerns:

1. I am puzzled why the authors fit the Ising model (and the mean field model) separately for every temperature. There are ways to include temperature directly into the model – for example, Ising models have an intrinsic dependence on (an abstracted) temperature, that is not included in equation 1 (line 157), where in typical formulations P(σ) = exp(-β*H(σ)) / Z, where β = 1/(k_b * T), so that the couplings J and biases h get naturally scaled by the temperature T, and the distribution over configurations σ becomes flatter with higher T. Of course the correspondence between the real ARTR and the real temperature, and the Ising model and the model temperature, should not be taken too literally. But qualitatively there may be an analogous effect, namely a reduction of effective couplings and a reduction in biases with increasing T, and a resulting flattening of the distribution over configurations (σ), producing a more "disordered" network with less coherence and shorter time constants. An alternative would be to treat temperature, or a function of temperature, as an input to the model, for example through the modified bias term h in the energy hamiltonian. Either of these seems more systematic than re-estimating the couplings and biases for every temperature, can the authors explain why they did it that way?

(I could not spot a systematic dependence on T of the parameters in Appendix 2 table 2 but as the authors explain, fitting the couplings and the biases can be hard to disentangle, however their comment "This owes to a progressive change in the values of both the couplings and the biases" suggests a systematic change, which I could not spot. However it could still work to incorporate a 1/T dependence explicitly.)

2. Then the traces in Figure 2A-D seem consistent with a loss of correlation amongst individual neurons in ARTR clusters, because the average becomes smaller (suggesting less coherence), but it is unclear because the Y units are "a.u.". We would need to know that the Y axis units in A is the same as that in B, and that the Y axis in C is the same as in D. Since the Y axes are labeled "a.u." so I am not sure whether this is the case. There is no reason not to provide true units (∆F/F for A,B, and average value for C,D) so that this comparison is possible.

3. Assuming my understanding of the above is correct, then, if the temperature dependence is incorporated in the model through for example the 1/T dependence or through the biases, it should be possible to fit the model at one temperature and then predict model behavior at all other temperatures. With this interpretation, the inclusion of multiple temperatures serves more as a "cross validation" of the model. I would find this a more convincing demonstration of the utility of the Ising model for network dynamics than what is currently demonstrated (i.e. more convincing than training on all temperatures and testing on all temperatures). This is analogous to the other strength of the model touted by the authors, namely that training on low-order statistics reproduces the higher-order statistics.

I realize that the temperature parameter T may not only reflect bath temperature, but also other factors like neuromodulatory inputs from the rest of the brain that depend on temperature, as the authors describe in the Discussion. Nevertheless, I think this should be attempted, even if the model T ends up scaling for example supralinearly with bath temperature, for example, T = bath-temperature + other-influences-that-are-a-function-of-T.

The same comments generalize to the mean field model. That is, I think it should be attempted to fit both the Ising and the mean field models to all temperatures and have the couplings and biases scale with 1/T (or 1/f(T) with f describing a combined dependence on bath temperature and other influences) or incorporate T into the biases as an external drive, instead of fitting the parameters separately for every temperature (of course individual fish should be fit separately). Then the held-out temperatures can serve for cross validation.

4. The way the Ising model is turned into something from which temporal dynamics can be read off is through the Monte Carlo sampler. This seems reasonable and elegant given what I understand about MC samplers and I trust it makes sense, but the paper explains the workings of the MC sampler poorly. There is a reference to some code online, but text providing the reader with a good intuition, or pseudocode, is missing. Given the importance of the MC sampler, the explanation of it, and why it can be interpreted as time evolution on long timescales, should be much more central.

5. Thus the interpretation of the Ising-ARTR correspondence seems to be first, that in both cases, less "coherence" between the units leads to longer time constants (under MC sampling in the model), and second, that increasing temperatures lead to less "coherence", thus linking temperature to time constants. If this is right, it would be useful to spell this out more clearly.

6. The Ising model is proposed to be a good model of ARTR network dynamics, and the model neurons (let's call these particles, for clarity, and call ARTR neurons, neurons) are fit to individual ARTR neurons. For readers, it would be very useful to see a graphical comparison of the dynamics of the particles, at the population level, and the dynamics of individual neurons, also at the population level. The Methods state that the mean and variance of each is fit but readers need a more personalized intuition for what the model looks like in comparison to the biological network. Although Appendix 2 Figure 1 and Appendix 2 Figures 2A(5) goes some way to depict the Ising particles and the real neurons, I think that having an elaborated population representation comparison figure in the main text would be useful, just for readers to visualize.

7. Line 165 "contralateral couplings vanish on average", also mentioned in the Discussion, seems strange because I would expect that contralateral couplings are on average negative. How does this fit in with contralateral inhibition? (As an aside, for biologists, the word vanish might be unclear and a different way of describing it might be better.) The discussion also says that the couplings are almost null but drive a subtle interplay. It seems to me that readers need to know if they are small but overall negative, or whether they can be small and overall positive; the latter would be hard to interpret but the former would be more straightforward to interpret. Why is the significance not quantified statistically?

8. The ARTR consists of excitatory and inhibitory clusters, as the authors describe, and refer to the Ising couplings, but I did not see a clear depiction or quantification of whether these couplings segregate, only a passing mention in the Discussion (lines 409-415). Can the authors depict the network connectivity accordingly, as particles with signed weights between them, and compared to their location in the brain?

9. In some ways, it is surprising that the behavior changes so much with temperature, because the field has learned from Eve Marder's group that network dynamics tend to be surprisingly robust to "crude" perturbations of the entire network. It is possible that it was difficult for evolution to compensate for the temperature dependence of ARTR, but it is also worth thinking about whether there might be an ethological purpose for the temperature dependence. Could it be useful for the animal to alternate turn direction more often when it's warm?

10. The phrasing "orientational persistence" came across as a bit of a misnomer to me, though I might be wrong – orientation typically means the angle of the animal in its environment, whereas this is about the change in angle, so orientation persistence might be interpreted by a casual reader as the fish swimming in a straight line at a fixed angle. I am not sure what else to call it though – "turn direction persistence" is probably more accurate but also clumsier. Is there a better term? If you decide to keep using "orientational persistence" it would be useful to very carefully and explicitly explain what you mean by it.

11. Cold-blooded animals have a body temperature that quickly converges to that of their surroundings (especially tiny fish). Their neural populations will likely experience a wider range of temperature fluctuation and therefore follow certain thermal dynamics laws unlike any warm-blooded animals (such that the flipping rate increases with temperature following Arrhenius law). It is important to discuss if the time persistence that emerges from the network model is a special case for cold-blooded animals experiencing different temperatures (through the 1/(kB*T) scaling of the couplings and biases), a direct thermal effect, rather than "purposeful" neural network computation (through external inputs that could be modeled by making the biases depend on T).

12. It is therefore good to discuss how applicable this free energy landscape description of a neural network is to other neural networks with persistent activities in other animal models, with different sensory inputs, in order to reach a larger audience and have wider potential applications. What types of time persistence can this interpretation address and what are its limitations?

13. Line 9: "networks" should probably be "network".

14. Figure 1 C and G: labeling the colors by their temperature in the figure would be helpful.

15. Line 246, it is Figure 3B showing the energy landscape, not 3C.

---

## [Author Response]

Essential revisions:(1) Please follow suggestions from reviewing on cross-validating model fitting(2) Fitting the model across temperature values and discussing the relationship between the physical and model temperatures(3) Analysis of temporal alignments between changes in the swim direction and the onset of sign change in the difference between the mean population activities of left and right hemispheres

We have followed these recommendations to write our amended manuscript. In particular, we now:

(1) include a systematic cross-validation of the inferred models following the comments by Reviewer 1. We show the predictions of the model (for observables such as the mean activity and the pairwise correlations, and for log-likelihoods) in the figures on the main text, e.g. figure 3A, as well as in the appendix figure 3.

(2) explain why we have fitted a model for each water temperature and each fish, and carry out a systematic comparison of the models inferred at different water temperatures for the same fish. We have added a new figure 3B to show the absence of correlation between models corresponding to different water temperatures. We have also included a new section in the Discussion, entitled “Origin and functional significance of the temperature dependence of the ARTR dynamics” to better discuss the role of temperature on the activity. We have also slightly rewritten the introduction of the model in the Results section to better distinguish the notions of water temperature and model temperature (implicitly set by the amplitude of the J,h parameters), as our first manuscript could be confusing on this point.

(3) point out the exact figure in Dunn et al., eLife 2016 in which the authors demonstrate that the sign of the difference in activity of the right and left ARTR populations dictates the swim direction. Replicating these observations would thus be redundant but would also require a different and more complex experimental setup. Indeed, they were obtained by recording fictive turns using electrical recordings from peripheral motor nerves in paralysed larvae, as actual tail bouts tend to occur at very low frequency in tethered configurations.

Reviewer #1 (Recommendations for the authors):1. Temporal alignment between neural activity and fish behavior.Line 101: "Impact of the bath temperature on the orientational persistence in freely swimming larvae".The authors compare the statistics of their sign(m_L-m_R) to the statistics of swim bouts in a previous paper and show that the two variables are highly correlated. However, this is just an indirect observation since they didn't establish whether changes flips in sign(m_L-m_R) are temporally aligned to changes in the fish swimming direction when analyzed simultaneously. I believe this is an important missing link in the manuscript and I suggest that the authors show that this relationship holds at least in one example – although it'd be ideal to show this at different temperature, it is sufficient to show it for one particular temperature. The dataset in Figure 4 is appropriate for this extra analysis, which should be performed both during spontaneous and stimulated epochs.Related to this issue: what is the probability of having a change in swim direction, without a flip of sign(m_L-m_R)? And vice versa, what is the probability of having a flip of sign(m_L-m_R) without a change in swim direction?

We agree with the reviewer that the relation between the change in swim turn direction and the neuronal switch between the left and right ARTR subpopulations, is central to the proposed scenario. This relationship has been previously established by Dunn et al. in their *eLife* article (Dunn 2016). Below we reproduce the panel E of figure 5 —figure supplement 2 from this article, which precisely demonstrates the robust correlation that exists between *m_L_-m_R_* and the turning direction. Notice that to obtain these results, the authors recorded fictive turns using electrical recordings from peripheral motor nerves on both sides of the tail of paralysed larvae, as actual tail bouts tend to occur at very low frequency in tethered configurations. In 2017, we also indirectly confirmed this relationship by establishing that *m_L_-m_R_* was a strong predictor of the gaze direction, and that the latter was strongly correlated with the tail bout orientations (Wolf et al. 2017).

We modified the manuscript to clarify this point by explicitly referring to this figure:

line 111-114

“It has previously been shown that the ARTR governs the selection of swim bout orientations: turn bouts are preferentially executed in the direction of the most active (right or left) ARTR subcircuit (Dunn et al., 2016, Wolf et al., 2017), such that sign (mL(t)−mR(t)) constitutes a robust predictor of the turning direction of the animal, see figure 5 —figure supplement 2E in (Dunn et al., 2016).”

Also related, what is the distribution of intervals between changes in the swim bout directions? If these behavioral observable is faster than the temporal resolution of the calcium indicator dynamics, this might introduce some statistical biases in the analysis.

The distributions of orientational persistence times, as measured experimentally, are shown in Author response image 1. Since two separate turning swim bouts are needed to detect a change in turn direction, this distribution shows a rapid decay for T<1s, which is the mean interbout interval. The left-right alternation process thus occurs on time scales that are particularly well adapted to calcium imaging, whose temporal resolution after spike inference is of the order of a few hundreds of ms.

**Author response image 1. sa2fig1:** 

2. Cross-validationLine 143: "Inference of an Ising model from functional recordings of the ARTR." The authors fit an Ising model to the data showing that it captures various observables in the data. However, the model is not cross-validated. The authors should use standard cross-validation procedures to establish the goodness of fit of the various observables on held-out data. In all plots in Figure 2C-F and Appendix 2 Figure 2B-E results should be replaced with a comparison with held-out data not used to fit the model.Moreover, what is the null hypothesis here? The authors should compare the held-out performance of their model to shuffled models trained on surrogate datasets obtained eg by destroying pairwise correlations.

In order to answer this important point, we include in the revised version of the paper a systematic cross-validation of the inferred models summarized in Appendix 2 Figure 3.

Indeed for each experiment, we divided the data sets in two parts: 75% of each data set is used as a training set and the remaining 25% is used as a test set. Each training set is used to infer an Ising model. We then compare the mean activity and covariance of the test set with the one computed from the simulated data generated by the models (see Appendix 2 Figure 3A-B). We also computed the relative variation of the models' log likelihood computed on the training data and the test data (see Appendix 2 Figure 3C). The very weak value of this relative variation confirms the quality of the fits.

Moreover, as suggested by the reviewer, we decided to use the independent model as a null hypothesis. We thus compared the Ising models fitted on the data with the independent model. The independent model only depends on the firing rates of the neurons. We then compare the mean activity and covariance of the test set with the one computed from the simulated data generated by the independent models (see Appendix 2 Figure 3E-G). As expected, these independent models with no connections poorly reproduce the data. The excess log likelihood confirms this result (Appendix 2 Figure 3G). Indeed, the relative variation, between the Ising and the independent models, of the log likelihood computed on the training data and the test data is around 50%.

In addition, we also followed the reviewer’s suggestion and computed the KL divergence in Figure 3 using the test sets. Indeed, the distribution of the KL divergence between the experimental test datasets and their associated Ising models is much smaller than those obtained between experimental test datasets and Ising models trained on different recordings (red distribution). We also added, for comparison, the distribution of KL divergence obtained using independent models, which demonstrate the importance of interneuronal connections to reproduce the data.

line 206–209

“This agreement crucially relies on the presence of inter-neuronal couplings in order to reproduce the pairwise correlations in the activity: a model with no connection (i.e. the independent model, see Methods) fitted to reproduce the neural firing rates, offers a very poor description of the data, see Figure 3A (dark blue distribution) and Appendix 2 Figure 3E-G.”

3. Definition of temperature in the modelIt is not explained at all how the water temperature appears in the model. Is it a missing 1/T factor in Equation (1)? This is a central issue the authors should be very clear about. What's a bit confusing is that the authors define "inverse temperature" as the effective size K, but there's no mention of the actual water temperature at all.Line 234-236: In the Langevin formalism, the diffusion constant D usually appears in the noise autocorrelation function = 2*D*δ(t-t'). This is usually taken to be equal to the temperature (inverse β) in the partition function (which is missing from its definition!). How are the statistical mechanical temperature and the actual water temperature entering here? This is very confusing, I strongly urge the authors to clarify this point.

We agree that the previous version of the manuscript was not clear enough in the definition and introduction of temperature(s). In the new version, we make clear that temperature refers to the temperature of water, and denote this quantity by T.

Regarding the Ising distribution defined in Equation (1), we implicitly set the model temperature T_model to 1, which amounts to omitting it in Equation (1). Considering a different value for T_model would simply amount to rescaling the biases and couplings parameters by the same quantity, and would not affect the model distribution.

The fact that the model temperature T_model is equal to unity is consistent with the distribution of left and right activities in Equation (3) and the Langevin dynamics in Equation (4), see covariance of the noise epsilon, which is 2* δ(t-t’). As stressed by the referee, the covariance should be equal to 2 * T_model * δ(t-t’), hence our expression corresponds to T_model=1.

We now clearly state after Equation 1 that the model temperature is set to one throughout the paper.

line 172–174

“Notice that in Equation 1, the energy term in the parenthesis is not scaled by a thermal energy as in the Maxwell–Boltzmann statistics. We thus implicitly fix the model temperature to unity; of course, this model temperature has no relation with the water temperature *T*.”

The use of the vocable inverse temperature (which was done with quotation marks) in Appendix right after Equation (18) to design the sizes K_L_, K_R_ was very confusing, and we apologize for that. We simply meant that, as the sizes K enter multiplicatively the free-energy F they could be formally interpreted as effective inverse temperatures in the Boltzmann factor e^-F^. We have removed this confusing part of the sentence, and hope that the current formulation is more clear.

Reviewer #2 (Recommendations for the authors):This is a beautiful study that combines physics with biology, treating a biological network as an inorganic system, by minimizing its free energy and maximizing entropy. It answers a lot of questions and hints at a valuable wider applicability of this approach. At the same time I am somewhat confused about the approach the authors have taken to fitting the models, explained below, and it would be good to have a conversation about that and possibly adjust some details of the approach, or to clarify. With appropriate revisions or clarifications this work would be a strong candidate for the journal.Comments, questions, suggestions, and concerns:1. I am puzzled why the authors fit the Ising model (and the mean field model) separately for every temperature. There are ways to include temperature directly into the model – for example, Ising models have an intrinsic dependence on (an abstracted) temperature, that is not included in equation 1 (line 157), where in typical formulations P(σ) = exp(-β*H(σ)) / Z, where β = 1/(k_b * T), so that the couplings J and biases h get naturally scaled by the temperature T, and the distribution over configurations σ becomes flatter with higher T. Of course the correspondence between the real ARTR and the real temperature, and the Ising model and the model temperature, should not be taken too literally. But qualitatively there may be an analogous effect, namely a reduction of effective couplings and a reduction in biases with increasing T, and a resulting flattening of the distribution over configurations (σ), producing a more "disordered" network with less coherence and shorter time constants. An alternative would be to treat temperature, or a function of temperature, as an input to the model, for example through the modified bias term h in the energy hamiltonian. Either of these seems more systematic than re-estimating the couplings and biases for every temperature, can the authors explain why they did it that way?(I could not spot a systematic dependence on T of the parameters in Appendix 2 table 2 but as the authors explain, fitting the couplings and the biases can be hard to disentangle, however their comment "This owes to a progressive change in the values of both the couplings and the biases" suggests a systematic change, which I could not spot. However it could still work to incorporate a 1/T dependence explicitly.)

In the new manuscript we have made clearer that β is a rescaling parameter, which is integrated in the inferred value of couplings, see answer to Major point 3 of Reviewer 1.

line 172–174

“Notice that in Equation 1, the energy term in the parenthesis is not scaled by a thermal energy as in the Maxwell–Boltzmann statistics. We thus implicitly fix the model temperature to unity; of course, this model temperature has no relation with the water temperature *T*.”

In addition, we now show in Appendix 2 Figure 6 some scatter plots of the couplings and biases at different temperatures for one fish, as well as statistics for all fish. These results clearly indicate the absence of any strong correlation between the Ising parameters at different temperatures (for the same animal). As a consequence, there is no simple global rescaling between the Ising parameters attached to two water temperatures. This is why we have fitted a model for each data set.

line 228–233

“We next examined the dependency of the Ising model parameters on the water temperature. To do so, for each fish, we selected two different water temperatures, and the corresponding sets of inferred biases and couplings, *{h_i_, J_ij_*}. We then computed the Pearson correlation coefficient *R^2^* of the biases and of the coupling matrices at these two temperatures (inset of Appendix 2 Figure 6). We saw no clear correlation between the model parameters at different temperatures, as shown by the distribution of *R^2^* computed across fish and across every temperatures (Appendix 2 Figure 6).”

2. Then the traces in Figure 2A-D seem consistent with a loss of correlation amongst individual neurons in ARTR clusters, because the average becomes smaller (suggesting less coherence), but it is unclear because the Y units are "a.u.". We would need to know that the Y axis units in A is the same as that in B, and that the Y axis in C is the same as in D. Since the Y axes are labeled "a.u." so I am not sure whether this is the case. There is no reason not to provide true units (∆F/F for A,B, and average value for C,D) so that this comparison is possible.

The notation a.u. was misleading, as we are showing the time trace of the mean (over each region) of the binarized activity. The axis labels have been corrected in Figure 2 (beware that the numbers of the panels have changed with respect to the first version of the manuscript).

3. Assuming my understanding of the above is correct, then, if the temperature dependence is incorporated in the model through for example the 1/T dependence or through the biases, it should be possible to fit the model at one temperature and then predict model behavior at all other temperatures. With this interpretation, the inclusion of multiple temperatures serves more as a "cross validation" of the model. I would find this a more convincing demonstration of the utility of the Ising model for network dynamics than what is currently demonstrated (i.e. more convincing than training on all temperatures and testing on all temperatures). This is analogous to the other strength of the model touted by the authors, namely that training on low-order statistics reproduces the higher-order statistics.

In the new manuscript, we include in the revised version of the paper a systematic cross-validation of the inferred models summarized in Appendix 2 Figure 3, see Point 2 in the Answer to Reviewer 1. Note that this cross validation is done at each temperature separately. We discuss the putative relationship between models at different temperatures below.

line 644–658

“We cross-validated the Ising models (see Appendix Figure 3) dividing the data sets in two parts: for each experiment, 75% of each data set is used as a training set and the remaining 25% is used as a test set. Each training set is used to infer an Ising model. We then compare the mean activity and covariance of the test set with the one computed from the simulated data generated by the models…” (see manuscript)

I realize that the temperature parameter T may not only reflect bath temperature, but also other factors like neuromodulatory inputs from the rest of the brain that depend on temperature, as the authors describe in the Discussion. Nevertheless, I think this should be attempted, even if the model T ends up scaling for example supralinearly with bath temperature, for example, T = bath-temperature + other-influences-that-are-a-function-of-T.The same comments generalize to the mean field model. That is, I think it should be attempted to fit both the Ising and the mean field models to all temperatures and have the couplings and biases scale with 1/T (or 1/f(T) with f describing a combined dependence on bath temperature and other influences) or incorporate T into the biases as an external drive, instead of fitting the parameters separately for every temperature (of course individual fish should be fit separately). Then the held-out temperatures can serve for cross validation.

Varying the water temperature was done here to investigate how the ARTR circuit persistent dynamics could be modeled across different behavioral conditions. The use of Ising is obviously not limited to this particular experimental protocol, which is of course adequate for cold-blooded animals only.

In general, we do not expect that the Ising models corresponding to different conditions can be related to each other in a simple way, e.g. through a global rescaling of the biases and/or of the couplings. However, we agree with Reviewer 2 that it is legitimate to attempt at relating the models inferred from the recordings at different water temperatures. We have done so in two ways:

– We have directly compared the couplings and biased inferred from the recordings of activity at different water temperatures for the same fish. Results are shown in Appendix 2 Figure 6, and show no clear interrelation between the parameters corresponding to different temperatures, as would be the case if parameters would transform according to a simple global rescaling.

– To confirm this finding, following the recommendations of Reviewer 2, we have carried out the following computation. We first infer the parameters h_i_ , J_ij_ of an Ising model from the recording of a fish at water temperature T. Then we rescale these parameters as a x h_i_ , b x J_ij_ and we generate synthetic activity with these rescaled ising models. We then compute the KL divergence between the distribution of activity defined by this rescaled Ising model and the recorded data at another temperature T’ (for the same fish). We then minimize this KL divergence over the global rescaling factors a and b in the [0.5;1.5] ranges, and compare this lowest divergence to the one between the same recorded data (at temperature T’) and the Ising model inferred from those data (as shown in Figure 3A of the main text). Ratios of these KL divergences are reported in Author response image 2. These results show that the global rescalings of the parameters inferred from data at temperature T cannot reproduce correctly the data at another temperature T’.

4. The way the Ising model is turned into something from which temporal dynamics can be read off is through the Monte Carlo sampler. This seems reasonable and elegant given what I understand about MC samplers and I trust it makes sense, but the paper explains the workings of the MC sampler poorly. There is a reference to some code online, but text providing the reader with a good intuition, or pseudocode, is missing. Given the importance of the MC sampler, the explanation of it, and why it can be interpreted as time evolution on long timescales, should be much more central.

We give now the details of the MC sampler in the main text and in methods of the revised version.

line 179-186

“The algorithm starts from a random configuration of activity, then picks up uniformly at random a neuron index, say, *i*. The activity *s_i_* of neuron *i* is then stochastically updated to 0 or to 1, with probabilities that depend on the current states *s_i_* of the other neurons (see Equation 8} in Methods, and code provided). The sampling procedure is iterated, ensuring convergence towards the distribution *P* in Equation 1. This *in silico* MC dynamics is not supposed to reproduce any realistic neural dynamics, except for the locality in the activity configuration **s** space.”

line 634-643

“In order to generate synthetic activity, we resorted to Gibbs sampling, a class of Monte Carlo Markov Chain method, also known as Glauber dynamics…” (see manuscript)

5. Thus the interpretation of the Ising-ARTR correspondence seems to be first, that in both cases, less "coherence" between the units leads to longer time constants (under MC sampling in the model), and second, that increasing temperatures lead to less "coherence", thus linking temperature to time constants. If this is right, it would be useful to spell this out more clearly.

The origin of the dependence of the persistence time on water temperature is rather subtle. We think that this dependence is not directly related to a change of the correlation structure in the activity. To support this point we provide a new panel (Appendix 2 figure 2C) showing that no significant relation between the average correlations and the temperature exists.

However, as our mean-field analysis shows, barriers increase with temperature, see Figure 4E, a fact that may appear counterintuitive at first sight. As a consequence, at high temperature, only the low-low activity state is accessible in practice to the system, and the mean activity remains low, see Appendix 2 Figure 2B, with fluctuations within the low-low state. Conversely, at low water temperatures, barriers separating the low-low and the high-low (or low-high) states are weaker, so the active states are accessible. This has two consequences. First, the mean activity is higher at low temperature (Appendix 2 Figure 2B). Second, the system may remain trapped for some time in such an active state before switching to the other side, and reversing activity. This is the origin of the longer persistence at low temperature.

We have improved the manuscript to make these points clearer, see end of section “Barriers in the free-energy landscape and dynamical paths between states”.

line 306-313

“As a consequence, at high temperature, only the low-low activity state is accessible in practice to the system, and the mean activity remains low, see Appendix 2 Figure 2D, with fluctuations within the low-low state. Conversely, at low water temperatures, barriers separating the low-low and the active high-low or low-high states are weaker, so the latter become accessible. As a first consequence, the mean activity is higher at low temperature (Appendix 2 Figure 2D). Furthermore, the system remains trapped for some time in such an active state before switching to the other side, e.g. from high-low to low-high. This is the origin of the longer persistence time observed at low temperature.”

6. The Ising model is proposed to be a good model of ARTR network dynamics, and the model neurons (let's call these particles, for clarity, and call ARTR neurons, neurons) are fit to individual ARTR neurons. For readers, it would be very useful to see a graphical comparison of the dynamics of the particles, at the population level, and the dynamics of individual neurons, also at the population level. The Methods state that the mean and variance of each is fit but readers need a more personalized intuition for what the model looks like in comparison to the biological network. Although Appendix 2 Figure 1 and Appendix 2 Figures 2A(5) goes some way to depict the Ising particles and the real neurons, I think that having an elaborated population representation comparison figure in the main text would be useful, just for readers to visualize.

We agree with Reviewer 2 that such a representation is useful. We now show in Figure 2 of the main text raster plots of the activity in the recording (panel A) and in the simulated Ising dynamics (panel C). We also provide in the supplementary materials the same comparison for every recording.

7. Line 165 "contralateral couplings vanish on average", also mentioned in the Discussion, seems strange because I would expect that contralateral couplings are on average negative. How does this fit in with contralateral inhibition? (As an aside, for biologists, the word vanish might be unclear and a different way of describing it might be better.) The discussion also says that the couplings are almost null but drive a subtle interplay. It seems to me that readers need to know if they are small but overall negative, or whether they can be small and overall positive; the latter would be hard to interpret but the former would be more straightforward to interpret. Why is the significance not quantified statistically?

We now provide a detailed analysis on both contralateral correlation and couplings in the Appendix 2 Figure 4. Indeed we show that strong negative correlation and couplings systematically correspond to pairs of contralateral neurons. We provide in the main text a paragraph on this point.

line 218-222

“… computing the fraction of neuronal pairs (*i,j*) that are contralateral for each value of the coupling ***J**_ij_* or the Pearson correlation (Appendix 2 Figure 4D-E). Large negative values of couplings or correlations systematically correspond to contralateral pairs of neurons, whereas large positive values correspond to ipsilateral pairs of neurons.”

8. The ARTR consists of excitatory and inhibitory clusters, as the authors describe, and refer to the Ising couplings, but I did not see a clear depiction or quantification of whether these couplings segregate, only a passing mention in the Discussion (lines 409-415). Can the authors depict the network connectivity accordingly, as particles with signed weights between them, and compared to their location in the brain?

We do not see a spatial organization in the inferred couplings that would reflect the neuronal type organization. We stress that the relationship between the inferred couplings ***J**_ij_* and the true physiological interactions is subtle. For instance, in our model, couplings are symmetric, which makes the distinction between inhibitory and excitatory neurons uneasy. In addition, the sampling time bin is long compared to synaptic integrations times, which makes causal dependencies hard to infer even with models where connections are not a priori required to be symmetric, see for instance Figure 7b and related discussion in a previous publication by two of us (SC,RM): J Comput Neurosci DOI 10.1007/s10827-010-0306-8

9. In some ways, it is surprising that the behavior changes so much with temperature, because the field has learned from Eve Marder's group that network dynamics tend to be surprisingly robust to "crude" perturbations of the entire network. It is possible that it was difficult for evolution to compensate for the temperature dependence of ARTR, but it is also worth thinking about whether there might be an ethological purpose for the temperature dependence. Could it be useful for the animal to alternate turn direction more often when it's warm?

The reviewer is right that some networks have evolved to maintain robust dynamics across a large thermal range. However, this is rather the exception than the rule, and even the frequency of the pyloric rhythm of the crab, studied by Marder, increases with temperature (although the phase is maintained).

Regarding whether or not the thermal dependence in ARTR dynamics is favorable for the animal, it is hard to conclude as many other kinematic features are also modified, as we have quantified recently. These different points are now discussed in a new section of the discussion.

line 379-400

“The brains of cold-blooded animals need to operate within the range of temperature that they experience in their natural habitat, e.g. 15-33°C for zebrafish [Gau et al., 2013]. This is a peculiarly stringent requirement since most biophysical processes are dependent on the temperature. In some rare instances, regulation mechanisms might stabilize the circuit dynamics in order to preserve its function, as best exemplified by the pyloric rhythm of the crab whose characteristic phase relationship is maintained over an extended temperature range [Tang et al., 2010]. Yet in general, an increase in temperature tends to increase the frequency of oscillatory processes [Robertson et al., 2012]. The observed acceleration of the ARTR left/right alternation with increasing temperature, could thus directly result from temperature-dependent cellular mechanisms. Furthermore, one cannot rule out the possibility that the ARTR dynamics could also be indirectly modulated by temperature via thermal-dependent descending neuromodulatory inputs.

As a result of this thermal modulation of the neuronal dynamics, many cold-blooded animals also exhibit temperature-dependence of their behavior [Long et al., 2008, Neumeister et al., 2000, Stevenson et al., 1990]. Here we were able to quantitatively relate the two processes (neuronal and motor) by demonstrating that an increase in temperature consistently alters the pattern of spontaneous navigation by increasing the left/right alternation frequency. Interpreting the functional relevance of this modification of the swimming pattern is tricky, since many other features of the animal’s navigation are concurrently impacted by a change in temperature, such as the bout frequency, turning rate, turn amplitude, etc. Nevertheless, we were able to show in a recent study that this thermal dependence of the swimming kinematic endows the larva with basic thermophobic capacity, thus efficiently protecting them from exposure to the hottest regions of their environment [Le Goc et al., 2021].”

10. The phrasing "orientational persistence" came across as a bit of a misnomer to me, though I might be wrong – orientation typically means the angle of the animal in its environment, whereas this is about the change in angle, so orientation persistence might be interpreted by a casual reader as the fish swimming in a straight line at a fixed angle. I am not sure what else to call it though – "turn direction persistence" is probably more accurate but also clumsier. Is there a better term? If you decide to keep using "orientational persistence" it would be useful to very carefully and explicitly explain what you mean by it.

We thank the reviewer for this suggestion: turn direction persistence is indeed better. We modified the text accordingly.

11. Cold-blooded animals have a body temperature that quickly converges to that of their surroundings (especially tiny fish). Their neural populations will likely experience a wider range of temperature fluctuation and therefore follow certain thermal dynamics laws unlike any warm-blooded animals (such that the flipping rate increases with temperature following Arrhenius law). It is important to discuss if the time persistence that emerges from the network model is a special case for cold-blooded animals experiencing different temperatures (through the 1/(kB*T) scaling of the couplings and biases), a direct thermal effect, rather than "purposeful" neural network computation (through external inputs that could be modeled by making the biases depend on T).

The observed speeding up of the ARTR alternating dynamics with the water temperature T is consistent with a direct thermal effect. We agree that the biases could also exhibit some variation with T. This is not unexpected, as different membrane currents may have different temperature dependence, see our response to point 9. However we do not see a strong dependence of the biases with T, see Author response image 3 showing the mean value of h_i_ as a function of the water temperature.

**Author response image 3. sa2fig3:** 

12. It is therefore good to discuss how applicable this free energy landscape description of a neural network is to other neural networks with persistent activities in other animal models, with different sensory inputs, in order to reach a larger audience and have wider potential applications. What types of time persistence can this interpretation address and what are its limitations?

Time persistence emerges in our work through the sampling of (generally) two metastable states. However, our approach could be applied to more complex landscapes, with more metastable states. Examples of such situations relevant for computation can be found in several works, e.g. Harvey et al. (2012), Mazzucato et al. (2019), Brinkman et al. (2022). It would be interesting to test our approach in the case of continuous attractors, e.g. ring attractors that serve as a theoretical basis for head direction networks (and their higher dimensional extension for place/grid cells circuits). Metastability in this context would be different as a continuum of states are expected to coexist. However we expect motion along the attractor direction, in particular drift, to happen on longer time scales than the damping of fluctuations along transverse directions. Some of us have already inferred graphical models on multi-electrode recordings of hippocampal place cells, see for instance Posani et al. (2018) PLoS Comp Bio 14:e1006320. However the limited number of recorded cells (compared to the recordings reported in our present manuscript) did not allow us to obtain sufficiently accurate generative performance with MC dynamics.

13. Line 9: "networks" should probably be "network".

This has been corrected.

14. Figure 1 C and G: labeling the colors by their temperature in the figure would be helpful.

The figure has been modified.

15. Line246, it is Figure 3B showing the energy landscape, not 3C.

This has been corrected.